# Acceptability of a nurse-led, person-centred, anticipatory care planning intervention for older people at risk of functional decline: A qualitative study

Dagmar A. S. Corry[1,2]*, Julie Doherty[1,2], Gillian Carter[1,2], Frank Doyle[3], Tom Fahey[3], Peter O'Halloran[1,2], Kieran McGlade[4], Emma Wallace[3,5], Kevin Brazil[1,2]*

1 Centre for Evidence and Social Innovation, Queen's University Belfast, Belfast, Northern Ireland, United Kingdom, 2 School of Nursing and Midwifery, Queen's University Belfast, Belfast, Northern Ireland, United Kingdom, 3 Department of General Practice, RCSI University of Medicine and Health Sciences, Dublin, Republic of Ireland, 4 School of Medicine, Dentistry, and Biomedical Sciences, Queen's University Belfast, Dunluce Health Centre, Belfast, Northern Ireland, United Kingdom, 5 Department of Health Psychology, Royal College of Surgeons in Ireland, Dublin, Republic of Ireland

* dagmar.corry@qub.ac.uk (DASC); k.brazil@qub.ac.uk (KB)

**Data Availability Statement:** The data underlying this study cannot be shared publicly because they are qualitative patient interviews which contain

## Abstract

### Background

As the population of older adults increases, the complexity of care required to support those who choose to remain in the community amplifies. Anticipatory Care Planning (ACP), through earlier identification of healthcare needs, is evidenced to improve quality of life, decrease aggressive interventions, and prolong life. With patient acceptability of growing importance in the design, implementation, and evaluation of healthcare interventions, this study reports on the acceptability of a primary care based ACP intervention on the island of Ireland.

### Methods

As part of the evaluation of a feasibility cluster randomized controlled trial (cRCT) testing an ACP intervention for older people at risk of functional decline, intervention participants [n = 34] were interviewed in their homes at 10-week follow-up to determine acceptability. The intervention consisted of home visits by specifically trained registered nurses who assessed participants' health, discussed their health goals and plans, and devised an anticipatory care plan in collaboration with participants' GPs and adjunct clinical pharmacist. Thematic analysis was employed to analyze interview data. The feasibility cRCT involved eight general practitioner (GP) practices as cluster sites, stratified by jurisdiction, four in Northern Ireland (NI) (two intervention, two control), and four in the Republic of Ireland (ROI) (two intervention, two control). Participants were assessed for risk of functional decline. A total of 34 patients received the intervention and 31 received usual care.

personal and potentially identifiable information, and participants have consented to publication of anonymous quotes only. Requests for data can be made to Queen's University Belfast Research Governance (contact via researchgovernance@qub.ac.uk) with appropriate ethical approval.

**Funding:** The Anticipatory Care Planning Study is funded from INTERREG VA funding of (incl. 15% contribution from the Department of Health in Northern Ireland and Republic of Ireland) that had been awarded to the HSC Research & Development Division of the Public Health Agency Northern Ireland and to the Health Research Board in Ireland for the Cross-border Healthcare Intervention Trials in Ireland Network (CHITIN) project (Grant number: CHI-5426; recipient: Prof. Kevin Brazil; sponsor: Queen's University Belfast: https://www.qub.ac.uk/). The funders had no role in study design, data collection and analysis, decision to publish, or preparation of the manuscript. The views and opinions expressed in this paper do not necessarily reflect those of the European Commission or the Special EU Programmes Body (SEUPB).

**Competing interests:** The authors have declared that no competing interests exist.

## Findings

Thematic analysis resulted in five main themes: timing of intervention, understanding of ACP, personality & individual differences, loneliness & social isolation, and views on health-care provision. These map across the Four Factor Model of Acceptability ('4FMA'), a newly developed conceptual framework comprising four components: intervention factors, personal factors, social support factors, and healthcare provision factors.

## Conclusion

Acceptability of this primary care based ACP intervention was high, with nurses' home visits, GP anchorage, multidisciplinary working, personalized approach, and active listening regarded as beneficial. Appropriate timing, and patient health education emerged as vital.

## 1. Introduction

High quality, personalized health care provision for older people at risk of functional decline remains the objective of health care systems in Northern Ireland (NI) and the Republic of Ireland (ROI) [1–3]. The United Kingdom National Health Service (NHS) and the Health Service Executive (HSE) in ROI strive to improve the quality of life for older people, enable them to retain their independence, and to live in their own homes for as long as possible. With an ever-increasing median age older people often live with multi-morbidities, making their long-term care more complex [4–6]. Staff shortages and wider systemic problems [7,8] result in reactive healthcare systems and the needs of older people often remain unmet [9–14], with inequalities of access to services.

Health systems in both jurisdictions strive to move from a medical care model towards a person-centred, holistic primary care model [15] with demonstrated benefits to patients' health [16]. It is widely acknowledged that preventative care models within primary care settings can reduce hospital and care home admissions and improve quality of life [17,18].

Person-centred care has been shown to improve patient experience, care quality, and health outcomes, including for those with long-term conditions [16–19], and to be more cost-effective [20]. Patients are less likely to use emergency hospital services, and more likely to adhere to their treatment and medication regimes when playing a collaborative role in their health and care [16]. Personalized, sensitive, and timely management of long-term conditions is key to facilitating this modern model of care. Anticipatory care planning ('ACP') has been defined as a process supporting those living with long term conditions to plan for an expected change in health or social status, incorporating health improvement and staying well [21]. ACP has a fundamental role to play in a person-centred, forward looking health care system in order to meet each patient's needs, respect their wishes and values, relieve their symptoms, and prevent or delay deterioration wherever possible. In ACP the patient is an active participant in their care planning rather than a mere recipient of care [16], and patients' perceptions as to whether such an intervention is acceptable to them will determine its success. Acceptability reflects the extent to which patients receiving a healthcare intervention consider it to be appropriate [22]. In the UK, the Medical Research Council [23] has significantly increased its references to acceptability in their guidance documents for evaluating complex interventions [24–26], indicative of the growing importance of this construct. Sekhon, et al. [22: p.5] defined acceptability as:

'. . . a multi-faceted construct that reflects the extent to which people delivering or receiving a healthcare intervention consider it to be appropriate, based on anticipated or experienced cognitive and emotional responses to the intervention.'

Against this background, the current paper aims to explore patient acceptability of a nurse-led, person-centred primary care ACP intervention for older adults at risk of functional decline on the island of Ireland.

## 2. Method

### 2.1 Design and procedure

This paper reports on patient acceptability of the intervention, and follows the COREQ guidelines for reporting qualitative research [27] (see S1 File) and the TIDieR Checklist [28] (see S2 File).

**2.1.1 The intervention.** The ACP intervention protocol has been described in detail elsewhere [29]. A feasibility cluster randomized controlled trial was conducted where, depending on the complexity of needs, those in the intervention group received up to three home visits (one to two hours in duration) over 10 weeks by specially trained registered nurses who assessed their physical, mental, and social health and discussed with them their health concerns, goals, and plans. Preceding the home visits, and to ensure consistency and a personalized care approach, registered nurses (n = 5) from both jurisdictions completed a three-day training programme designed to orientate them to the intervention and study procedures. The training was facilitated by a clinician expert in the field, and the programme included study overview, principles and practice of personalized care, shared decision making, conduction of a standardized, person-centred, holistic assessment with the EASY-Care [30] tool, and completing a medication review in collaboration with a clinical pharmacist. The ACP assessment using EASY-Care was conducted with the aid of a medical summary provided by the GP practice, including details of the patients' health conditions and prescribed medications. In consultation with participants' GPs and an adjunct clinical pharmacist who conducted the medication review the nurses developed a personalized care plan supported by the GP, and this was then shared with the patient.

The RE-AIM conceptual framework guided the evaluation of the ACP intervention [19,31]. Under the 'Adoption' component within this framework thematic analysis [32] was employed to explore the acceptability of the nurse-led ACP intervention for older adults at risk of functional decline. This involved qualitative interviews with participants in their own homes at 10-week follow-up (August to October 2019) following completion of the intervention. During the visit, quantitative data was also collected. Participants completed quantitative questionnaires for the 10-week follow-up, then the qualitative interview. Family carers could take part in the interviews at the participant's discretion. The qualitative interviews had a median average length of nine minutes (range: three to 24 minutes), were audio recorded and transcribed verbatim, and were accompanied by field notes. The interviews were conducted by an experienced female researcher (DC) who had not met the participants prior to interview.

### 2.2 Sample

Eight general practitioner (GP) practices were assigned as cluster sites to either the intervention or control arm (four per group). Practices were stratified by jurisdiction, and further by rurality prior to randomisation. GP database systems were searched to identify eligible participants and a chart audit and PRISMA-7 screening form [25] used to screen for risk of functional decline for inclusion. The inclusion criteria were aged 70+; two or more chronic

medical conditions; four or more regularly prescribed medications; a PRISMA-7 score of $\geq 3$; a hospital admission in the previous year; three or more physician visits in the past year, and the ability to complete an English language questionnaire. Full details can be found in the protocol paper [29]. PRISMA-7 is considered a best-practice tool to identify patients at risk of frailty in general practice, with those obtaining a score of $\geq 3$ recognized as being at increased risk [33,34].

Out of 73 patients meeting eligibility, 65 were recruited and randomly allocated to intervention (n = 34) or control (n = 29) group after consent and baseline data collection. All patients in the intervention group were invited to complete a qualitative interview at 10-week follow-up (August to October, 2019) to explore the acceptability of the ACP intervention.

### 2.3 Interview schedule

A semi-structured interview schedule guide was developed consisting of questions pertaining to patient acceptability (appropriateness, benefits, and convenience) of the intervention. The schedule was informed by the RE-AIM framework [31,35], and questions were based on a review of related research and the expertise of the research team, including patient and public involvement (PPI). Following GRIPP guidelines on PPI reporting [36] we engaged three PPI (one in ROI, two in NI) in an advisory capacity to attend regular project team meetings and to discuss progression, next steps, and consult on study documents, including qualitative and quantitative interview schedules, to ensure the vital lay person perspective is incorporated. An example of PPI input to the qualitative interview schedule is the change from 'Did you feel actively involved in your discussions with the nurse to identify your healthcare needs?' to 'Did you have enough input in identifying your health needs and developing your care plan?' Interview questions assessed patient perceptions of the intervention including the overall intervention, its component parts (patient meetings, assessment, patient education on anticipatory care planning), implementation (was the home environment suitable for meetings, were the contents reviewed in meetings helpful) and suggestions for improvements to the intervention. The interview schedule guided discussions and when necessary, prompts were used. All interviews took part between August and October 2019. Topic guide items included the following:

- 'What did you expect from the care planning exercise before meeting with the nurse for the first time?'

- 'How was the overall process of taking part in the care planning exercise?'

- 'Did you have enough input in identifying your health needs and developing your care plan?'

- 'What is the value of completing an anticipatory care plan?'

- 'Did your taking part in the study help your life in any way?'

### 2.4 Ethical considerations

Ethical approval was obtained in the ROI from the Research Ethics Committee, Irish College of General Practitioners in January 2019 (reference ICGP2018.4.10). In NI approval was received from the Office for Research Ethics, Northern Ireland (reference 19/NI/0001). Following ethical approval and prior to individual baseline data collection visits between February and June 2019, all participants provided written, informed consent to participate in, and be interviewed about, the study.

## 2.5 Data analysis

NVivo-12 was used to help organize and manage the data. The lead author thematically analyzed the transcribed interview data [28] in an inductive approach, in collaboration with another member of the research and writing team (KB). The intervention nurses were not part of the writing team. We created an open and modifiable codebook, sought, identified, and interpreted patterns, commonalities and differences, leading to a theme structure and final thematic framework. We used data triangulation (interviews, notes, and observation), source triangulation (participants from two jurisdictions) as well as researcher triangulation (two researchers involved in data analysis) in order to strengthen our findings and improve rigour. Researchers observed reflexivity to minimise potential bias and influence. Pseudonyms (IDs) were used; IDs ending in NI denote participants from NI; those ending in ROI denote participants from the ROI. IDs beginning with L and F indicate urbanicity, while those beginning with M and E show rurality.

# 3. Findings

There was no attrition (n = 34; ROI = 19 (55.9%); NI = 15 (44.1%), with all of the intervention participants agreeing to be interviewed. Table 1 below provides a brief summary of intervention participant characteristics; Table 2 in (S3 File) offers a detailed overview.

The average PRISMA-7 score of 4.15 (1.12) in our sample was indicative of an increased risk of frailty and the need for further clinical review. As per inclusion criteria all participants had two or more chronic conditions; were taking on average 11.39 medications; had 5.2 GP visits during the past year; and an average of 6.4 inpatient nights in the previous year. Only one family carer actively took part in the interview.

All participants were white European. Gender was evenly distributed, with a mean sample age of 80.13. The majority were married (61.8%) and lived in urban areas (58.82%).

Analysis of interview data resulted in five main themes: Timing of intervention, understanding of ACP, personality & individual differences, social isolation, and views on healthcare provision. Based on these themes we developed an overarching conceptual framework of patient acceptability onto which these five main themes mapped. The new framework comprises four interacting components: intervention factors, personal factors, social support factors, and healthcare provision factors, and forms the basis of the Four Factor Model of Acceptability ('4FMA'). The 4FMA assists the evaluation of patient acceptability by recognising its multifactorial nature, and identifying facilitators and barriers. The four components of the model, along with their respective main themes are illustrated in Fig 1 below. Details of our findings in each of those components and their main themes are presented in the following.

## 3.1 Intervention factors

The overarching component of 'intervention factors' on the 4FMA includes aspects of the intervention which facilitated or hindered acceptability. Appropriate timing was key to ensure acceptability. Lack of understanding of ACP was a barrier, indicating that improving health literacy should be a priority. The home visits, the psychosocial aspect of the nurses' visits, and their ability to actively listen and build rapport were facilitators, as were the practical support they provided and the pharmacist's medication review.

**3.1.1 Timing of intervention.** Many participants did not want to contemplate a less able future and felt they were 'not there yet'. They believed that the questions posed to them as part of the intervention were perhaps not all relevant as they felt physically and mentally quite well although some discussed their fear of deteriorating health and how the intervention helped them face this.

**Table 1. Intervention participant characteristics summary.**

| Characteristics | Details/Inclusion Criterion (IC) | Mean (SD) | Number (%) |
|---|---|---|---|
| **Gender distribution** | Female | - | n = 16 (47.06%) |
| | Male | | n = 18 (52.94%) |
| **Age: 70+ years** | IC | M = 80.13 (5.70) | n = 34 (100%) |
| **Prisma-7 score ≥3** | IC | M = 4.15 (1.12) | n = 34 (100%) |
| **Marital status & living arrangements** | Married | - | n = 21 (61.8%) |
| | Living with partner | | n = 13 (38.2%) |
| | Living with adult children | | n = 8 (23.5%) |
| | Living alone | | n = 13 (38.2%) |
| | • Widowed | | n = 8 (23.5%) |
| | • Divorced | | n = 3 (8.8%) |
| | • Single | | n = 2 (5.9%) |
| **Rural/Urban Distribution** | Rural | - | n = 14 (41.18%) |
| | Urban | | n = 20 (58.82%) |
| **Medications** | IC | M = 11.03 (3.59) | n = 34 (100%) |
| **Number of hospital inpatient nights in previous year** | IC | M = 6.4 (26.5) | n = 34 (100%) |
| **Physician visits in previous year** | IC | M = 5.2 (3.4) | n = 34 (100%) |
| **Ability to complete an English language questionnaire** | IC | - | N = 34 (100%) |
| **Family carer participation** | female (spousal) carer | - | n = 1 (2.94%) |

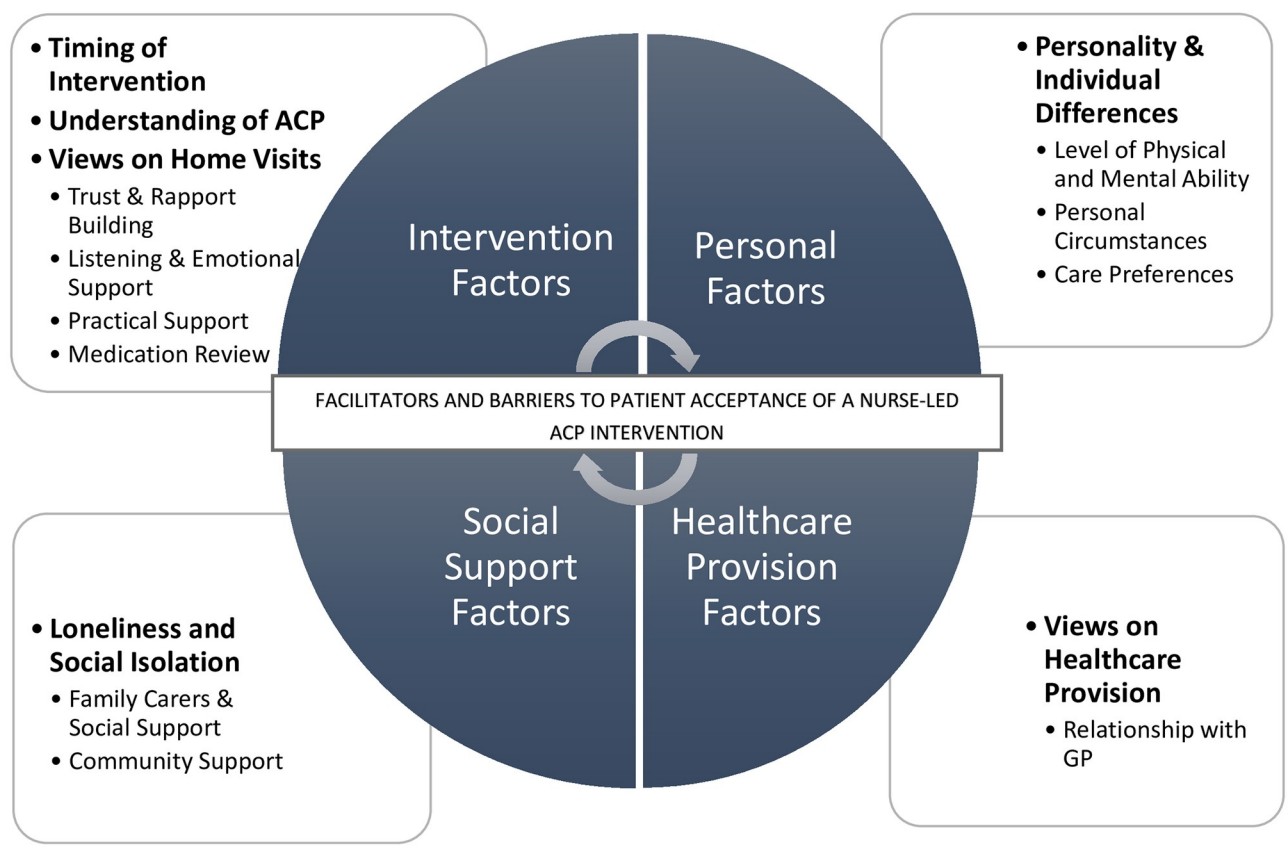

**Fig 1. Four factor model of patient acceptability of nurse-led anticipatory care planning intervention ('4FMA').**

'It's made me think about a lot of things that I wasn't trying to think about, and didn't want to think about, and it's like facing your fears. If you face your fears, you are not afraid of them then.'

(L011NI).

While some did not regard the intervention as appropriate for them at this point in time, 'It's hard to judge what we have in front of us. I couldn't possibly judge that.' (M004NI), others were able to identify immediate benefits.

'I got the opportunity to say I may need help with stairs or what not in a few years' time. Otherwise, I wouldn't have had that opportunity to say that, and I am sure I am not the only one that would be in that position.'

(L010NI).

**3.1.2 Understanding of ACP.** The intervention was viewed favourably by participants, as attested by the affirming feedback during interviews, illustrated by the quotes throughout the findings. However, some considered themselves to be in relatively good health and, therefore, believed it was perhaps not immediately beneficial to them.

'Tomorrow, next week, six months' time, it might be far more relevant for me, and at the present minute I would have said it wasn't terribly relevant.'

(M004NI).

In contrast, others found the intervention appropriate and beneficial, valuing the information provided and knowledge gained.

'I think it helped because it gives you an idea of what can happen in the future and what help you *can* get you know. The nurses explained a whole lot of that too by saying about, as you get older you might need a care package or you might need help, so I think it was very informative and it was helpful.'

(L005NI).

**3.1.3 Views on home visits.** The home visits were well received by all participants, with high praise for the nurses, and several participants expressing hope to receive regular such nurses' visits over a longer period to ensure continued monitoring and support.

'What I would like for somebody to, say, every so often just keep a check on me.'

[F91087ROI].

Participants benefitted from the information, and the practical and emotional support provided, acknowledging the appropriateness of the intervention.

'They recognized things that I was going through, the loss of my friend, they made great suggestions about what I could do.'

(L011NI).

**3.1.4 Trust and rapport building.** The nurses' proficient, person-centred attitude was of great importance for patient acceptability of the intervention as a whole, and the home visits in particular.

'They [nurses] were both professional, and well-meaning, and intelligent people so I felt quite at ease with them.'

(M008NI).

Indicative of the nurses' ability to build rapport and trust, participants reported they made them feel comfortable and relaxed, and they appreciated the effort to come and see them in their homes as it was more convenient for them.

'You feel more at home in your own surroundings, maybe you are more likely to tell them things that you might not discuss in more formal surroundings.'

(M008NI).

Indeed, for some it would have been challenging to meet elsewhere due to mobility problems.

**3.1.5 Listening and emotional support.** The appropriateness of active and compassionate listening became very clear, and many participants said that the most beneficial aspect of the intervention for them was the caring and personal contact. They felt reassured and less 'forgotten about' (M006NI). Home visits averaged 90 minutes' duration which allowed for a relaxed approach and time to build rapport and offer personalized support. 'They spent time with me. I didn't feel rushed.' (L011NI). Participants felt safe to talk about emotional difficulties as well as physical conditions, appreciating the benefits and convenience of a whole-person approach.

'I found it very informative, they were friendly, and I felt as if somebody actually cared and it was very reassuring. And I think they helped me a lot because I talked to them and I felt free to talk to them. And in fact they cared. Sometimes when you get older you feel people don't care about you anymore and you are useless, and you know, you are just a bother. But it was the opposite of that.'

(L011NI).

**3.1.6 Practical support.** Participants reported having derived a range of practical benefits from the nurses' visits, including dietary changes (E32854ROI), exercise advice (E54137ROI), guidance with personal arrangements, e.g., making a will (E69601ROI), a Do-Not-Resuscitate order (E69601ROI, F91087ROI), assistance with tax-free home adaptations (L005NI), and social prescribing (F44050ROI).

'First of all: I made a will, I didn't do that before the nurse came.'

(E69601ROI).

**3.1.7 Medication review.** The convenience of the medication review was recognised; indeed, some participants regarded the review as the most impactful element of the intervention.

'And one thing that struck me actually which I think was positive, that list of medications I've given you; I was pleased that part of this was having that impartially looked at and

reviewed because I could imagine that there is a danger as these things gradually go on, over a period of years, so to know that there was a pharmacist, looking impartially at that, was something I was pleased about, and relieved about; it wasn't a major worry I had before but when I thought about it, that's very useful.'

(L002NI).

## 3.2 Personal factors

The overarching component 'Personal factors' on the 4FMA provided insight in terms of 'personality & individual differences' on the acceptability of the ACP intervention. Within that main theme, the sub-themes of 'level of physical and mental ability', 'personal circumstances', 'care preferences' and 'financial aspects' emerged as facilitators and barriers respectively.

**3.2.1 Personality & individual differences.**   Personality factors seemed to impact on acceptability. Some participants were keen to present themselves as able and independent, yet their beliefs and behaviour associated with their health and future prospects varied. They reported keeping active and feeling well, taking a pro-active approach to maintaining their health, and believing that they could maintain their current level of health well into the future despite their comorbidities, and therefore were reluctant to accept that they needed an intervention at this point in time.

'It's difficult to know what's in the future, you'd need a crystal ball, you know. I think at first I would try to keep as active as possible and would almost think like be very positive so, I think now ok, people can have strokes and heart attacks, but at the moment I don't see that happening and I do try and keep reasonably healthy. I enjoy life, so.'

(L010NI).

Others appeared more resigned to their physical decline, perhaps less pro-active in promoting their own health, and focusing on the deficits of their situation. They welcomed the intervention with open arms.

'Just if you could put in for a nurse coming, say, every three or four months to pop in and see how things are 'cause over a few months things can change, especially at my age. On the 19th of this month, I am 82, so from then on you never know what's round the corner.'

(F91087ROI).

**3.2.2 Level of physical and mental ability.**   In addition to a variety of physical conditions, there was a wide range of mental agility observed in participants, ranging from very lucid and animated to hard to engage in meaningful exchange, and lack of focus. The latter was sometimes linked to e.g., hearing impairments and at other times seemingly an aspect of personality and individual circumstances. For some, impaired hearing exacerbated difficulties in communicating and relating, as did memory loss (F91087ROI). Participants who were aware of the complex and progressive nature of their conditions and understood that they would need help in the future to maintain their independence and quality of life, and potentially extend their life expectancy, readily acknowledged its benefits.

'Well, it would be important to me if it kept me living and kept me moving, that's the main thing that I would worry about if—as long as I knew that somebody was there to help.'

(L001NI).

Others, with readily available informal care, felt they were not currently disadvantaged by their condition/s, seemed unable or unwilling to identify with a frail cohort, and so did not regard ACP of immediate importance for them at that point in time.

'I imagine that my physical condition is probably mild compared to some of the people that you have to deal with so I feel as though I am a bit of a fraud doing this because I don't have real serious health problems that a lot of people do.'

(M008NI).

**3.2.3 Personal circumstances.** Having the opportunity to discuss fears and worries as part of the intervention was perceived as beneficial, particularly by those living alone and feeling isolated. Participants' personal circumstances varied considerably and many were keen to share their stories, predominantly regarding personal loss, bereavement, regrets, family worries, relationship issues, and concerns over potential health deterioration. The intervention gave them the opportunity to do so while alleviating fears about an unknown future.

'I feel more reassured about it now. I was frightened about the future but now I've discussed it and talked about it, I'm not as afraid of it now as I was.'

(L011NI).

**3.2.4 Care preferences.** When asked about their understanding about their future care needs, several participants expressed a strong aversion to the prospect of having to go into a care home, with some fearing financial loss or abuse.

'The only thing I am fearful of is being put into a home and losing your money. I fear that. I say to [daughter] every now and again, I'm not going to no home, you know.'

(L003NI).

Confirming the appropriateness, convenience, and benefits of the ACP intervention, a clear preference for being cared for at home was expressed.

'Say if you were confined to be—or all these kinds of things, you would rather have help in your home.'

(L005NI).

'My mother and father both had to go into nursing homes, not nursing homes but residential homes, and I hope I never have to. Nobody knows.'

(M004NI).

## 3.3 Social support factors

The overarching component 'social support factors' on the 4FMA provided insight in terms of the influence of 'loneliness & isolation' on the acceptability of the ACP intervention. This main theme contains the sub-themes of 'family carer', 'friends & family', and 'community support'.

Many participants reported good social support which appeared synonymous with greater life satisfaction and perceived better health, and therefore did not recognise the value of ACP

to them at this point in time (e.g., L005NI, L007NI, L010NI, E38659ROI). Conversely, some felt ignored and uncared for and found the intervention very appropriate and beneficial.

'Because as you get older and that, you just feel, you're gonna be in the ground in another while, and who would bother or care about us, d'you know that sort of attitude that nowadays they...treat old people, as [partner] says about the doctor, it's age, if your time's up they forget about you then. Well, you've served society and its, you know, time to move on.'

(E54137ROI).

**3.3.1 Loneliness and isolation.**    The intervention appeared to be particularly acceptable to patients who were lonely and isolated. This included but was not exclusive to those living in rural areas. Being divorced or widowed (n = 11, 32.3%), living alone (n = 13, 38.2%), very old age and a high level of frailty were all reasons for isolation and loneliness. Illness impacting on mobility and the inability to leave their house unaided stopped participants from engaging in activities they had previously enjoyed.

'As I said before...how will I say it...my social life has gone. I have to stay in when I would love to go out and do different things, and I have to stay in because of my head.'

(F82139ROI).

Having impaired hearing was another contributing factor to social isolation making it very difficult to engage in conversations and meaningful social interaction. In all those instances, the nurses' home visits were considered very convenient, and the intervention highly beneficial as participants felt listened to, reassured, cared for, and less isolated.

'I found the whole thing very helpful because of, it makes you feel that you are not isolated or forgotten. That somebody is actually thinking, we'll see if this person could get help, so I think it's a sort of a reassurance, that's what I felt.'

(L005NI).

**3.3.2 Family carers & social support.**    The acceptability of the intervention was influenced by the level and quality of informal care, and social support currently received by participants. The better cared for someone felt, the less they thought the intervention was relevant to them, and vice versa. The majority of participants received informal care (n = 21, 61.8%), with nine being cared for by their spouse, 11 by adult children, and one by extended family, whom they were dependent on for daily tasks and for company. However, informal care was often difficult, with some family carers having additional caring commitments alongside being sole carer for their family member, e.g., adult children who had their own families and work obligations to consider (e.g., F73211ROI, E53448ROI) and struggled considerably with the added burden. Equally, some participants had caring obligations themselves. One, whose husband also took part in the study; and who cared for her son with mental health problems, found the information and actionable advice she received during the study highly beneficial to all of them.

'I feel that, maybe getting the respite for [husband] was a start, and then [son] getting sorted, you know that he is much better, too, you know.'

(E59405ROI).

Despite often having health problems of their own, spousal family carers routinely provided some or all support (e.g., L004NI, L009NI, L002NI, L010NI, E69601ROI, E36988ROI, E39713ROI), leaving participants worrying what would happen if and when their spouse becomes unable to care for them due to illness.

'While my wife's living and we are together and, other than my back, we are both in fairly good health—I like to think we are—but that's today, who knows next week?'

(LB010NI).

The intervention was regarded as particularly appropriate and beneficial by participants who lived alone, and devoid of a family carer experienced a lack of care, a sense of isolation, and uncertainty regarding care when their health deteriorates further.

'I think for some people who are more isolated they feel that somebody has taken an interest and will follow it up with the doctor, I think psychologically it would be very important to have that, and very supportive, and ah, they know then that's how it's been, this has been made known.'

(L002NI).

Those with good support from family and friends saw somewhat less benefit in the ACP intervention. Participants derived a sense of safety, joy, and belonging from having extended family and good friends to support them (e.g., L007NI, L005NI, E39603ROI, E84283ROI, E34839ROI, F75177ROI, F82139ROI), adding to an overall sense of wellbeing and a reluctance to acknowledge the appropriateness of a forward planning health intervention.

'I honestly don't really see, looking into the future, whatever it may hold, I don't think I'm going to really need any other support than what I've already got.'

(L007NI).

'I'd be on the skype there to my son or my daughter and my brothers. It's like being in the same room with them. If I want a bit of company, I go in next door and I give [name omitted] a shout and we might go off for a few pints.'

(F91087ROI).

**3.3.3 Community support.** Only a few participants responded to social prescribing during the intervention but where this was the case it was regarded as highly beneficial.

'It was just going to a club and that was a great idea. It's brilliant, its somewhere to go different you know and a bus collects us and all; it's brilliant you know.'

(F44050ROI).

A number of participants reported having good neighbourly connections, and a supportive community, while others were isolated and lonely due to personal circumstances, including living alone, living rurally, and being of an introverted disposition. Some attended community groups e.g., Men's Shed (E49587ROI), or were members of organizations, e.g., Parkinson's Society (L002NI) which afforded them awareness and knowledge about where to turn for help

if required. Having an existing social support network appeared to foster the belief that an ACP intervention was not immediately appropriate.

'I think probably because we were reasonably well informed through the Parkinson Society, maybe the overall impact [of the intervention] for us would not be as marked as for someone who's more isolated.'

(L002NI).

Group membership did not suit everyone, however, and not being able to drive or walk longer distances impacted on participants' ability to uphold their social connections. The subsequent perception of being isolated contributed to a feeling of being 'forgotten about', leading to high acceptability of the intervention. Those who lived alone, with little help and few connections found the intervention very beneficial in terms of alleviating social isolation, providing reassurance that they are cared for, and that care is available to them.

'*What did you find was the most helpful part of taking part in the study*?' 'Well you might laugh at me when I say this—the company. The company coming in. Yeah, that I wasn't forgotten.'

(F82139ROI).

### 3.4 Healthcare provision factors

The overarching component 'healthcare provision factors' on the 4FMA provided insight in terms of the impact of participant 'views on healthcare provision' on the acceptability of the ACP intervention. The main theme contained the sub-theme of 'relationship with GP'. Participants expressed satisfaction at their healthcare generally.

'I can't complain really about the health service. I know people do, but there's nothing perfect; but as far as I am concerned . . . I can't complain, for everything possible has been done for me.'

(L007NI).

**3.4.1 Views on healthcare provision.** Participants were aware of the high demand on GP surgeries and related primary care services, leading to excessive waiting times; and frequently reported not wanting to 'bother' the doctor. Participants whose conditions were relatively mild, with little need for frequent healthcare appointments, attributed less importance to the intervention than those with complex comorbidities who already needed frequent appointments. The latter readily perceived the benefits and convenience of a trained nurse coming to their home and assisting with current and future health needs. Some were aware of ongoing challenges within the health service and, in light of this, the appropriateness of the intervention.

'The bottom line is that the doctors' surgeries you know they are overrun anyway, if anything can be done to improve mobility, the movement of people in and out, that seems to be a jam, you could spend, sit 3.5 hours, and that's the only part that I would like to see improved, you know and if this is a sort of an exercise that will go towards alleviating that you know, I'd be on for that.'

(E53448ROI).

Inequality in access to services due to rurality meant an even greater need for, and appreciation of, the benefits of the home based, nurse-led ACP intervention.

'When you get to 70 you have to think about things, when you can't drive, what are you gonna do, I mean I couldn't really live in this area, because there is no buses, so that's something that you have to consider, but if you get a bit of help from the health service that you can maintain your independence then that makes a big difference.'

(M008NI).

The benefits of having a trusted health professional visiting participants in their own home, providing holistic care with the view to safeguarding independence, were recognized and welcomed.

'Well, I am hoping I'll be able to keep as independent as I possibly can with advice and help, maybe from a nurse calling in occasionally just to check up on me and see if I'm alright.'

(L011NI).

**3.4.2 Relationship with GP.** GP involvement in the intervention was key as participants trusted their GP which facilitated recruitment and encouraged participation. In fact, some expected their GP to be involved as a matter of course.

'I mean I would have been surprised if the GP hadn't been involved because they are there in overall control of your health.'

(M008NI).

Despite trusting their GP participants reported avoiding making appointments where possible as they did not want to 'bother' them, potentially disappearing from the GP's radar and not receiving the care they need. One participant provided insight into the reality of some people not wanting to 'bother' their doctor and how the intervention may benefit them.

'I think it would be very useful for people to have this service in the future. I think it would be very useful if they could have a sort of person that would come along to people over a certain age and give them that reassurance. Talk to them about their health, talk to them about their mental feelings, and reassure them, and maybe advise them to go to their doctor and get more help. Some people when they get older they don't want to bother anybody, they don't want to go to the doctor, they don't want to bother the doctor. But if you have somebody coming to you, who is a trained person who knows what they are doing and how to do it, it makes it so much easier.'

(L011NI).

Finally, no substantial differences in participant acceptability of the intervention were observed between the two jurisdictions, indicating trans-jurisdictional transferability and representativeness of the findings.

## 4. Discussion

### 4.1 Overview of findings

We sought to elicit user perceptions on the appropriateness, convenience, and benefits of the ACP intervention through qualitative interviews as per the 'Adoption' component of the 'RE-AIM evaluative framework [31,35]. We found that patient acceptability of the ACP intervention was high, but depended on multiple factors. The newly established 4FMA has emerged from the findings and provides an evaluation framework for patient acceptability in terms of intervention and patient inherent facilitators and barriers. Four overarching components comprising intervention factors, personal factors, social support factors, and healthcare provision factors facilitated or hindered patient acceptability. In line with existing literature [15,16] what transpired is that multidisciplinary working [37] and a personalized approach [17,38] were key to the success of the ACP intervention. The intervention overall, its primary care setting with GP anchorage [5,38], home delivery by a specially trained nurse, and involvement of a pharmacist were considered both appropriate and beneficial. Some patients believed that the timing was not right for them, indicating perhaps a need for patient health education to ensure their understanding of the trajectory of their complex conditions and the value of ACP in light of this [39,40]. The home visits were regarded as convenient, and the medication review, psychosocial aspects of the nurses' visits, and practical help provided were perceived as beneficial.

Notwithstanding the advantages of improved access to practical help and advice (e.g., dietary changes, exercise advice, guidance with making a will, a Do-Not-Resuscitate order, assistance with tax-free home adaptations, and social prescribing), and medication review, participants reported deriving great benefit from the psychosocial aspect of the nurses' visits, their active listening, compassion, and personal validation. Psychosocial support concerns in older adults are often a feature of chronic physical health and social support issues [41–46]. This was particularly true for those in our sample who felt lonely, isolated and 'forgotten about, which notably included the hearing impaired [47].

Preventive home visits are potentially beneficial models of comprehensive care, reducing mortality and care-home admissions for frail, older adults [48]. Unanimously, participants valued the nurses' home visits, as they felt comfortable and safe in their own environment. Importantly, experienced nurses in the project quickly built trust and established rapport, which enabled participants to speak openly about their physical, mental, and social difficulties. This was essential to both the holistic assessment and patient acceptability. Active listening, showing compassion, spending appropriate time with patients, helping to advise and make choices contributes to building a trusting respectful relationship [49]. Gaining patients' trust is associated with acceptance of, and adherence to, recommendations, lower anxiety, accessing services, participant autonomy, and shared decision making [33].

GP involvement as facilitator and anchor for the intervention has proved crucial as participants unanimously held their GP in high esteem and trusted their judgement in terms of participation. Treatment acceptability has been suggested to refer to individual components of an intervention [34,35,37], with social acceptability denoting the appropriateness of the intervention considering individual differences [43–45]. There was sometimes a reluctance to identify with a frail cohort in our sample. While some participants fully appreciated the intervention, others considered themselves to be still reasonably healthy and coping well, albeit with the support of their family carer. Therefore, they believed the ACP intervention was not immediately relevant to them. Perceived relevance depended on level of physical and mental ability, understanding of the intervention, individual differences–including personality, social support, personal, and financial circumstances—and insight into future trajectories of conditions. This brought into focus the importance of appropriate timing of the ACP intervention,

personalisation, and patient health education. Health education for older adults can be very effective, both in terms of improving intervention adherence and potentially in reducing morbidity and excess mortality [40,50]. It could help improve health literacy [39,42] and ensure knowledge and understanding of ACP, thus facilitating timely uptake.

## 4.2 Implications

Based on the high acceptability of multidisciplinary working with GP anchorage, including a specially trained nurse and a pharmacist as the main stakeholders alongside the patients, there are some clear implications deriving from the findings of this feasibility cRCT. To render this primary care intervention feasible and acceptable it would require an allocated, specially trained ACP nurse and adjunct pharmacist, with direct access to other health and social care professionals. This would facilitate multidisciplinary working, improve access and patient outcomes. The approach should be patient–centred, with well-timed holistic assessment and treatment.

In terms of implications for future research, a full trial of the ACP intervention should take on board the feedback provided by participants in terms of acceptability. This means retaining those components which worked well (GP anchorage, home visits by a specially trained nurse, holistic assessment, medication review, person-centred approach, multidisciplinary working), while improving the timing of the intervention, and including health education in order to increase understanding of the intervention and to manage expectations. Health education for older adults can be very effective, both in terms of improving intervention adherence and potentially in reducing morbidity and excess mortality [40,50]. It could help improve health literacy [39,42] and ensure knowledge and understanding of ACP, thus facilitating timely uptake.

## 4.3 Strengths and limitations

The study's strengths include provision of rich data on the participant acceptability of the ACP intervention, with zero attrition in the intervention group. Findings show an absence of cross-jurisdictional differences, indicating their transferability to other ethnically white populations. The data-driven (bottom-up) Four Factor Model of Acceptability ('4FMA') has been developed which could be applied to similar studies. While it shares some elements with Sekhon et al.'s [25] Theoretical Framework of Acceptability ('TFA'), such as the TFA's affective attitude and experience, which map onto the 4FMA's intervention factors, it goes beyond that to include social support factors, healthcare provision factors, and personal factors. The model recognises these four factors, and their interaction as facilitators and barriers respectively for acceptability of the intervention. The study adds to research in participant acceptability of healthcare interventions.

As the sample was entirely ethnically white transferability to other ethnic groups may be limited. An unexpected limitation was that, despite an average PRISMA-7 score of 4.15, some participants perceived themselves 'too well' to fully benefit from the intervention indicating that initial assessment and selection criteria may warrant modification to ensure that timing of the intervention is appropriate for participants. The findings also underlined the importance of expectation management from the outset. Participants should receive health education as to the purpose and processes of ACP, and both should be clearly and continuously communicated to ensure participants know what they can expect and in which timeframe.

## 5. Conclusion

This primary care ACP intervention as a whole found unanimous acceptance in our sample, as did its individual components. The multidisciplinary approach through the collaboration of

GP, nurse, and pharmacist provided the bedrock of the intervention. GP anchorage was key to successful recruitment and acceptability. Home visits by the trained nurse were enthusiastically received and perceived as very convenient and helpful, with socially isolated patients particularly welcoming the psychosocial aspect of the visits. The person-centred approach taken by the nurses was crucial to rapport building, holistic assessment, and acceptability. The medication review provided by the adjunct pharmacist was recognized as being a very useful aspect of the intervention. The timing of the intervention requires careful thought, and the inclusion of health education for patients is advisable. The newly developed 4FMA could be applied to similar patient acceptability studies. This ACP intervention has the potential to future-proof the management of complex, multiple conditions and improve quality of life for older adults, enabling them to live in their own home as independently as possible.

## Supporting information

**S1 File. Consolidated Criteria for Reporting Qualitative Studies (COREQ): 32-Item checklist.**
(DOCX)

**S2 File. The TIDieR (Template for Intervention Description and Replication) checklist.**
(DOCX)

**S3 File. Participants characteristics details.**
(DOCX)

## Acknowledgments

The authors wish to acknowledge the invaluable contribution of our participants, GPs and practice managers, and express our gratitude to them for giving so generously of their time. We thank the Northern Ireland Clinical Research Network [Primary Care] (NICRN PC) for the recruitment of GP practices and the delivery of the ACP intervention in Northern Ireland.

## Author Contributions

**Conceptualization:** Dagmar A. S. Corry, Julie Doherty, Gillian Carter, Frank Doyle, Emma Wallace, Kevin Brazil.

**Data curation:** Dagmar A. S. Corry.

**Formal analysis:** Dagmar A. S. Corry.

**Funding acquisition:** Kevin Brazil.

**Investigation:** Dagmar A. S. Corry.

**Methodology:** Dagmar A. S. Corry, Julie Doherty, Gillian Carter, Frank Doyle, Tom Fahey, Peter O'Halloran, Kieran McGlade, Emma Wallace, Kevin Brazil.

**Supervision:** Kevin Brazil.

**Validation:** Julie Doherty, Gillian Carter, Frank Doyle, Tom Fahey, Peter O'Halloran, Kieran McGlade, Emma Wallace.

**Writing – original draft:** Dagmar A. S. Corry.

**Writing – review & editing:** Dagmar A. S. Corry, Julie Doherty, Gillian Carter, Frank Doyle, Tom Fahey, Peter O'Halloran, Kieran McGlade, Emma Wallace, Kevin Brazil.

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
