## [Decision Letter · Decision Letter 0]

17 Nov 2020

PONE-D-20-30516

Acceptability of a nurse-led, person-centred, anticipatory care planning intervention for older people at risk of functional decline: A qualitative study

PLOS ONE

Dear Dr. Corry, 

Thank you for submitting your manuscript to PLOS ONE. After careful consideration, we feel that it has merit but does not fully meet PLOS ONE’s publication criteria as it currently stands. Therefore, we invite you to submit a revised version of the manuscript that addresses the points raised during the review process. 

Please respond to the editor and peer-review and comments detailed below.

We look forward to receiving your revised manuscript.

Kind regards,

Catherine J Evans, PhD, MSc, BSc (Hons)

Academic Editor

PLOS ONE

Journal Requirements:

Additional Editor Comments:

Editor comments

Please to take care with formatting and proof reading and use of acronyms ROI and NI - please state respective country names in full for international readers to define acronyms when first used line 14 .....island of Ireland, Northern Ireland (NI) and the Republic of Ireland (ROI).

Please improve the clarity of the reporting in the methods

The methods state paper follows COREQ guidelines. Please include as a supplementary file your completed COREQ checklist for your paper and indicate in the manuscript (see supplementary file 1)

Please give detail on the intervention - the only detail given is the number of visits, that conducted by a research nurse who assessed health domains and liaised with GP and pharmacist to devise an anticipatory care plan. This is too brief for a complex intervention. Please review to the TIDier checklist for intervention description and replication. How was the research nurse trained, was the intervention manualized for consistency, did this incorporate any evidence-based interventions to e.g. medicine optimization i.e. STOPPfrail. If your population was older people with frailty, were the frailty syndromes considered in the assessment e.g continence, falls, reduced mobility. Or a general 'health assessment' - what informed the components. if this is detailed in the published trial protocol, please give sufficient detail of the intervention for the reader to understand what it comprised. How was this assessment 'person-centred' - this is the main message in the conclusion, yet no detail is given as to how this was person-cetnred. We need understanding on the detail in the intervention that person-centred and how e.g. asked what are the person's priorities, used an evidence-based tool e.g. person-centred comprehensive PCOM i.e. ESAS/IPOS, or Staying Well Check tool https://www.england.nhs.uk/wp-content/uploads/2017/11/dg-case-study-staying-well-check-tool.pdf

Did the same research nurse deliver the intervention and conduct the qualitative interviews? Please can you clarify in your methods. This is important for the rigour of your qualitative study. Please state in your methods when the qualitative interviews were undertaken in relation to the feasibility trial , e.g. after completion of the intervention at XX weeks post randomisation. Please also state if the interviews were conducted in a separate data collection time point or with the quant data. This is can be brief if detailed in the protocol. But this is important to understand the quality of your embedded qualitative study in the feasibility trial.

Please remove reporting on randomisation and sampling from the intervention detail. Randomisation and sampling could move to your sampling section.

In the methods 2.3 sample section - please move reporting on your sample to the results, and detail your method of sampling in the methods. The sampling states eligibility criteria for the trial. How were the participants for individual interviews purposively selected. Please state the criteria used.

Section 3 Findings

Please begin with the detail reporting the participants moving this detail from the methods section. Please provide more detail on your sample - mean/SD or median and IQR for PRISMA with interpretation e.g. moderate frailty; mean/median age and respective measures of dispersion, ethnicity; and illness factors e.g. main diagnosis group by ICD-10 chapter headings i.e. cardiovascular disease. or if multiple conditions state to give some sense of why considered this patient group likely to benefit from ACP . This data is important to understand your sample and applicability of your findings for other populations. I would suggest putting this demographic data in table 1: Participant characteristics. Also please detail how many interviews was a caregiver present. And please state how representative is your qualitative sample from the feasibility trial sample. How many people approached for the qualitative interview agreed to participate, what were the reasons for decline. Your qual work indicates that participants considered themselves 'too well' for the ACP, and those who were frailer indicated greater benefit. Please considered if the patients who agreed for interview were generally 'well' and more able to participate in this aspect of the study causing potential sampling bias.

Discussion

This is a qualitative study. Please use language describing your results aligned with qualitative methodology. For example line 496 - very effective, is quant positivist language. please use nouns/adjectives to report qual findings e.g. essential, crucial. Please review the discussion and structure to tighten the reporting on the main messages made. Key points seems the 'holistic' assessment undertaken by the research nurse and sense of being 'heard and valued', multidisciplinary team working particularly intervention of the pharmacist for medicine review/optimization. The role of the GP from your findings would seem more than 'appreciated' (line 580). Important to give a sense of how the intervention worked from the findings.

The RE-AIM framework is reported in the first para of the methods, but there is no further reference as to how RE_AIM was applied with the study e.g. used to inform design of the topic guide, data analysis, data interpreting? If RE-AIM not used in this aspect of the study please remove from the methods, or detail in your methods how applied and pick this up in the discussion. For example - How does RE_AIM relate to your 4FMA framework identified in your data analysis?

Conclusions - these need to be tightened to reflect your findings.

Reviewers' comments:

Reviewer's Responses to Questions

**Comments to the Author**

1. Is the manuscript technically sound, and do the data support the conclusions?

Reviewer #1: Partly

Reviewer #2: No

2. Has the statistical analysis been performed appropriately and rigorously? 

Reviewer #1: N/A

Reviewer #2: Yes

3. Have the authors made all data underlying the findings in their manuscript fully available?

Reviewer #1: No

Reviewer #2: Yes

4. Is the manuscript presented in an intelligible fashion and written in standard English?

Reviewer #1: No

Reviewer #2: Yes

5. Review Comments to the Author

Reviewer #1: Thank you for the opportunity to review this manuscript. It is a very interesting topic and good to see implementation acceptability being reported for ACP. Please find following my general, and more specific comments which I feel will make this a stronger manuscript.

Overall:

Background: You explain the background in great depth and there are some very relevant points. However, there are some statements which do not appear strictly relevant to building your argument. Reducing some of these peripheral statements would make this section stronger and improve readability.

Quotes: Throughout you change formatting between quotes indented and quotes within text. It is off-putting for the reader. Decide which and stick with it. Very short quotes (a few words) within the text are fine.

Discussion: This feels quite long and would benefit from greater contextualisation in the wider literature.

Implications: Again this feels long. The manuscript would benefit from the repetitions being cut and this section focussing on potential clinical and research impacts. You also mention health education here for the first time. Please introduce this in the Discussion and strengthen your argument with external literature. Overall you mention education five times, including in the conclusion, but this is not then mentioned in your abstract conclusions. If this is a significant implication it should be noted in the abstract.

Conclusions: This could be more clearly linked to your key messages.

Formatting: As PLOS ONE does not copyedit accepted manuscripts please do check over the full manuscript and figure for typographical errors such as use of semicolons and capitalisation etc.

Specific:

Line 10: I would have liked to see the main aim reflected here.

Lines 17-18: This is rather confusing. Is that four practices in the control and four intervention? Or four patients within each of the eight practices to control and four patients within each of the eight practices to intervention? Neither adds up to the n=34 you interviewed. I make 8 practices and 8 participants at each to be a total of 64, half of which, the intervention group, would be 32. Obviously you know how many people you interviewed so I must have misunderstood something in the way the numbers are explained. This needs clarifying throughout the manuscript.

Line 25-26: I do not understand the relevance of the hyphens here or later in the manuscript (e.g. Line 111)

Line 26-28: These are not articulated the same as the main headings in your findings section (Lines 161-163). Please correct.

Line 98: Can you add that participants gave informed consent here? I appreciate you say it below, but not to see it here raises a red flag to the reader.

Lines 105-106: See Lines 17-18 comment above.

Line 109: Move the bracketed element after 'for inclusion'. Also not sure you need the brackets, just a separate sentence. It would make this clearer that there is more data if the reader wishes.

Line 110: How many met the inclusion criteria? Why did you decide on only 8 from each practice? What is your rationale?

Line 112: What was the dose? Did some only receive one? Why didn't everyone receive the same dose? Specifying would help the reader decide the weight of the findings.

Line 115: Specify this was the nurse. 'They' can be confusing and imply it was the team of GP, pharmacist and nurse. If it was the team, this needs to be clarified throughout as previously it has implied it was the nurse that constructed the plan.

Line 121: Why face to face?

Line 131: Can you reference some of this literature?

Line 132: Did you use PPI? If not, it may be useful to explain your justification for this.

Line 144: When? Also January 2019?

Line 148: Who conducted the data analysis. All members of the research team? Is the research team the writing team?

Line 165: Make it clearer that this is a conceptual framework which you have developed and give more details. Also, potentially mention as an output in your abstract.

Line 172-176: Describing what is in the chart doesn't add anything. Try and give a high level summary for this theme here.

Line 194: Where is the evidence for this?

Line 226: Impossible is a very strong word. Challenging? Difficult?

Line 268-271: I find this statement very uncomfortable. How did you analyse negative and positive mindsets? I appreciate participants gave what appeared to be two different views. Maybe report on that rather than your interpretation of what mindset that meant they had.

Line 277: This is the first mention of frailty. Are you diagnosing frailty as a syndrome, if so, how, or saying some patients were more susceptible to functional decline? I would change this title or quantify. Perhaps Impact of (fluctuating?) mental and physical capacity?

Line 281: Personality. Can you explain further? Is this your analysis from the interview? If these are participants who are living with frailty, this could merely be a be a fluctuation rather than a person's personality.

Line 282: Was capacity one of your inclusion criteria? If not did that impact the intervention and your study? You do not mention this in your limitations or elsewhere.

Line 283-284: This sentence is confusing to me. (The following section makes perfect sense.) I am unsure what this togetherness of physical and mental capacity means, or where your evidence is to clarify it.

Line 324: The use of the word subsequent makes the sentence somewhat confusing.

Line 337 and 505: Again, please clarify what you mean by the term frailty in this context.

Line 357: Is spouse the right word when your example is regarding adult children?

Line 366-368: Can you evidence this with a quote?

Reviewer #2: I think this is a useful study to examine the effects of nurse-led Anticipatory care planning.

I did not understand the following points, so I would like to ask for additional explanation.

・In the Intervention part, it is stated that the GP was divided into the intervention group and the control group by Feasibility RCT, but if this study is a cluster RCT design, please describe the flow chart of the participants. If it's not a cluster RCT design, shouldn't this description be included in the sample?

In addition, although it is divided into an intervention group and a control group, please describe the correspondence to the control group.

・If this study is cluster RCT design, I think the sample size is small.

・Please specify the meaning and effect of collecting data separately for the intervention group and the control group.

6. PLOS authors have the option to publish the peer review history of their article (what does this mean?). If published, this will include your full peer review and any attached files.

Reviewer #1: No

Reviewer #2: No

---

## [Author Response · Author response to Decision Letter 0]

1 Feb 2021

Response to Reviewers

The authors would like to express their sincere gratitude to the editors and reviewers for giving of their valuable time to review our submission and providing their helpful suggestions for revision, all of which we have taken on board, and endeavoured to implement and incorporate, thus making it a stronger manuscript. We have addressed each point raised during the review and responded to it with the relevant remedial action and response as indicated below.

Points raised by Editors and Reviewers Authors’ response and action

1. Please ensure that your manuscript meets PLOS ONE’s style requirements, including those for file naming. The PLOS ONE style templates can be found at

https://journals.plos.org/plosone/s/submission-guidelines

THE AUTHORS HAVE ENDEAVOURED TO ENSURE THAT THE MANUSCRIPT MEETS PLOS ONE’S STYLE REQUIREMENTS.

2. We note that you have indicated that data from this study are available upon request. 

PLOS only allows data to be available upon request if there are legal or ethical restrictions on sharing data publicly. For information on unacceptable data access restrictions, please see http://journals.plos.org/plosone/s/data-availability#loc-unacceptable-data-access-restrictions.

OUR COVER LETTER WILL NOW INDICATE THAT WE HAVE SUBMITTED OUR ANONYMISED INTERVIEW DATA AS SUPPLEMENTARY FILES. 

3. (Point 3 is a repeat of point 2.) 

Additional Editor Comments:

Please to take care with formatting and proof reading and use of acronyms ROI and NI - please state respective country names in full for international readers to define acronyms when first used in line 14 … island of Ireland, northern Ireland (NI) and the Republic of Ireland (ROI). THE AUTHORS HAVE FOLLOWED THE PLOS FORMATTING GUIDELINES, AND HAVE CAREFULLY PROOF-READ THE MANUSCRIPT.

THE ABSTRACT NOW DISPLAYS COUNTRY NAMES IN FULL, FOLLOWED BY THEIR ACRONYMS AT FIRST MENTION IN LINES 21 AND 22:

THE FEASIBILITY CRCT INVOLVED EIGHT GENERAL PRACTITIONER (GP) PRACTICES AS CLUSTER SITES, STRATIFIED BY JURISDICTION, FOUR IN NORTHERN IRELAND (NI) (TWO INTERVENTION, TWO CONTROL), AND FOUR IN THE REPUBLIC OF IRELAND (ROI) (TWO INTERVENTION, TWO CONTROL). ‘ISLAND OF IRELAND’ DOES NOT USE AN ACRONYM. 

Please improve the clarity of the reporting in the methods. THE AUTHORS HAVE ENDEAVOURED TO IMPROVE THE CLARITY OF THE REPORTING IN THE METHOD SECTION.

The methods state paper follows COREQ guidelines. Please include as a supplementary file your completed COREQ checklist for your paper and indicate in the manuscript (see supplementary file 1). THE COMPLETED COREQ CHECKLIST IS NOW INCLUDED AS A SUPPLEMENTARY FILE (S3) AND THIS IS INDICATED IN TEXT (PAGE 5).

Please give detail on the intervention - the only detail given is the number of visits, that conducted by a research nurse who assessed health domains and liaised with GP and pharmacist to devise an anticipatory care plan. This is too brief for a complex intervention. Please review to the TIDier checklist for intervention description and replication. MORE DETAIL HAS NOW BEEN PROVIDED ON THE INTERVENTION IN SECTION 2.2 IN THE METHODS, IN KEEPING WITH THE TIDieR Checklist, WHICH HAS BEEN COMPLETED AND IS ALSO PROVIDED AS A SUPPLEMENTARY FILE (S4), INDICATED ON PAGE 5.

How was the research nurse trained: RESEARCH NURSES COMPLETED A THREE-DAY TRAINING PROGRAMME DESIGNED TO ORIENT THEM TO THE INTERVENTION AND STUDY PROCEDURES. THE TRAINING WAS FACILITATED BY A CLINICIAN EXPERT IN THE FIELD. THIS IS NOW STATED ON PAGE 6. 

Was the intervention manualized for consistency, did this incorporate any evidence-based interventions to e.g. medicine optimization i.e. STOPPfrail. If your population was older people with frailty, were the frailty syndromes considered in the assessment e.g., continence, falls, reduced mobility. Or a general 'health assessment' - what informed the components. if this is detailed in the published trial protocol, please give sufficient detail of the intervention for the reader to understand what it comprised. YES, THE INTERVENTION WAS MANUALISED WHICH IS REFLECTED IN THE TRAINING OF THE NURSES IN TERMS OF STANDARDISED ASSESSMENT, REPORTING, AND ENGAGEMENT WITH GP. THE MEDICATION REVIEW WAS BASED ON BEST EVIDENCE. THE ACP ASSESSMENT WAS CONDUCTED USING THE STANDARDISED EASY-CARE TOOL (Philip KE, Alizad V, Oates A, Donki DB, Pitsillides C, Syddal SP, et al. Development of an EASY-Care, for brief standardized assessment of the health and care need of older people; with latest information about cross-national acceptability. J Am Med Dir Assoc. 2014;15: 42-6) WHICH INVOLVES SEVERAL DOMAINS TO ENSURE A PERSONALISED HOLISTIC APPROACH. THIS WAS SUPPLEMENTED WITH A MEDICATION REVIEW. PAGE 6 NOW PROVIDES THIS INFORMATION.

How was this assessment 'person-centred' - this is the main message in the conclusion, yet no detail is given as to how this was person-centered. We need understanding on the detail in the intervention that person-centred and how e.g. asked what are the person's priorities, used an evidence-based tool e.g. person-centred comprehensive PCOM i.e. ESAS/IPOS, or Staying Well Check tool https://www.england.nhs.uk/wp-content/uploads/2017/11/dg-case-study-staying-well-check-tool.pdf

PAGES 5-6 NOW INCLUDE: THE RESEARCH NURSES (5) WERE TRAINED BY AN EXPERT CLINICIAN. TO ENSURE A PERSONALIZED CARE APPROACH, REGISTERED NURSES FROM BOTH JURISDICTIONS COMPLETED A THREE-DAY TRAINING PROGRAMME INCLUDING STUDY OVERVIEW, PRINCIPLES AND PRACTICE OF PERSONALIZED CARE, SHARED DECISION MAKING, CONDUCTION OF A HOLISTIC ASSESSMENT WITH THE EASY-CARE TOOL, AND COMPLETING A MEDICATION REVIEW IN COLLABORATION WITH A CLINICAL PHARMACIST. 

How was intervention person-centred: THE STUDY AIMS, TRAINING PROVIDED TO THE RESEARCH NURSES, AND THE EASY-CARE ASSESSMENT TOOL ALL ALIGNED TO REFLECT PERSON-CENTRED CARE. THIS IS NOW BETTER EXPLAINED IN THE INTERVENTION SECTION, PAGES 5-7.

Did the same research nurse deliver the intervention and conduct the qualitative interviews? Please can you clarify in your methods. This is important for the rigour of your qualitative study. THIS HAS NOW BEEN CHANGED. PAGE 5 STATES THAT THE INTERVIEWS WERE CONDUCTED BY AN EXPERIENCED RESEARCHER. PAGES 5-6 STATE THAT THE INTERVENTION WAS CONDUCTED BY SPECIALLY TRAINED RESEARCH NURSES.

Please state in your methods when the qualitative interviews were undertaken in relation to the feasibility trial, e.g. after completion of the intervention at XX weeks post randomisation. Section 2.1 PAGE 5 STATES THAT THE QUALITATIVE INTERVIEWS WITH PARTICIPANTS TOOK PART AT 10-WEEK FOLLOW-UP (AUGUST TO OCTOBER 2019) IMMEDIATELY AFTER COMPLETION OF THE INTERVENTION AND AT THE SAME TIME AS TIME 2 QUANTITATIVE DATA COLLECTION.

Please also state if the interviews were conducted in a separate data collection time point or with the quant data. This is can be brief if detailed in the protocol. But this is important to understand the quality of your embedded qualitative study in the feasibility trial. WE HAVE NOW INDICATED THAT THE QUALITATIVE INTERVIEW TOOK PLACE AT THE SAME TIME AS TIME 2 QUANTATIVE DATA COLLECTION. SECTION 2.1 PAGE 5.

PLease remove reporting on randomisation and sampling from the intervention detail. Randomisation and sampling could move to your sampling section. THIS HAS BEEN MOVED TO THE SAMPLE SECTION.

In the methods 2.3 sample section - please move reporting on your sample to the results, and detail your method of sampling in the methods. THIS HAS BEEN DONE.

The sampling states eligibility criteria for the trial. How were the participants for individual interviews purposively selected. Please state the criteria used. ALL INTERVENTION PATIENTS WERE INTERVIEWED; WE HAVE NOW OMITTED THE WORD ‘PURPOSIVELY’. (PAGE 7).

Section 3 Findings

Please begin with the detail reporting the participants moving this detail from the methods section. THIS HAS BEEN DONE.

Please provide more detail on your sample - mean/SD or median and IQR for PRISMA with interpretation e.g. moderate frailty; mean/median age and respective measures of dispersion, ethnicity; and illness factors e.g. main diagnosis group by ICD-10 chapter headings i.e. cardiovascular disease. or if multiple conditions state to give some sense of why considered this patient group likely to benefit from ACP. This data is important to understand your sample and applicability of your findings for other populations. IN LINE WITH UK NATIONAL HEALTH SERVICE RECOMMENDATIONS THE BEST PRACTICE PRISMA-7 SCREENING TOOL WAS APPLIED TO IDENTIFY PATIENTS AT RISK OF FUNCTIONAL DECLINE / FRAILTY. PRISMA-7 IDENTIFIES THOSE WITH A SCORE OF THREE OR HIGHER AS AT INCREASED RISK FOR FRAILTY. THIS HAS NOW BEEN EXPLAINED IN THE ‘SAMPLE’ SECTION. MEANS AND SD FOR PRISMA SCORES AND FOR AGE HAVE NOW BEEN INCLUDED IN THE FINDINGS SECTION, AS HAVE GENDER AND ETHNICITY. AS PER INCLUSION CRITERIA ALL PARTICIPANTS HAD TWO OR MORE CHRONIC CONDITIONS.

THE SUITABILITY OF, AND RATIONALE FOR, THE CHOSEN PATIENT GROUP IS PROVIDED IN THE INTRODUCTION. 

I would suggest putting this demographic data in Table 1: Participant characteristics. THIS TABLE (TABLE 1: PARTICIPANT CHARACTERISTICS) IS NOW INCLUDED (PAGE 7).

Also please detail how many interviews was a caregiver present. ONE CARE PARTNER ONLY WAS PRESENT AT INTERVIEW. THIS HAS NOW BEEN STATED IN THE FINDINGS. 

And please state how representative is your qualitative sample from the feasibility trial sample. How many people approached for the qualitative interview agreed to participate, what were the reasons for decline. ALL INTERVENTION PARTICIPANTS WERE INVITED AND AGREED TO TAKE PART IN THE INTERVIEWS. THIS IS NOW STATED IN THE FINDINGS TOO. 

Your qual work indicates that participants considered themselves 'too well' for the ACP, and those who were frailer indicated greater benefit. Please considered if the patients who agreed for interview were generally 'well' and more able to participate in this aspect of the study causing potential sampling bias. ALL INTERVENTION PARTICIPANTS (n=34) AGREED TO PARTICIPATE IN THE QUALITATIVE INTERVIEW THUS AVOIDING SAMPLING BIAS. THIS IS NOW STATED IN THE ‘SAMPLE’ SECTION. 

Discussion

This is a qualitative study. Please use language describing your results aligned with qualitative methodology. For example line 496 - very effective, is quant positivist language. please use nouns/adjectives to report qual findings e.g. essential, crucial. WE HAVE ENDEAVOURED TO REMOVE SPECIFICALLY QUANTITATIVE LANGUAGE.

Please review the discussion and structure to tighten the reporting on the main messages made. Key points seems the 'holistic' assessment undertaken by the research nurse and sense of being 'heard and valued', multidisciplinary team working particularly intervention of the pharmacist for medicine review/optimization. The role of the GP from your findings would seem more than 'appreciated' (line 580). Important to give a sense of how the intervention worked from the findings. WE HAVE REVIEWED AND STRUCTURED THE DISCUSSION, FOCUSING ON THE MAIN POINTS FROM THE FINDINGS. 

GP ANCHORAGE HAS INDEED BEEN IDENTIFIED AS ESSENTIAL RATHER THAN ‘APPRECIATED’. THIS HAS BEEN AMENDED ACCORDINGLY.

The RE-AIM framework is reported in the first para of the methods, but there is no further reference as to how RE_AIM was applied with the study e.g. used to inform design of the topic guide, data analysis, data interpreting? RE-AIM, MORE SPECIFICALLY THE ‘ADOPTION’ COMPONENT (WE HAVE CLARIFIED THAT NOW IN ‘DESIGN AND PROCEDURES’), WERE USED AS A FRAMEWORK TO GUIDE THE EVALUATION, AND AS SUCH GUIDED THE TOPIC GUIDE (AS STATED IN THE SECTION ‘INTERVIEW SCHEDULE’ OF THE METHODS), AND DATA ANALYSIS (STATED IN THE ‘DESIGN AND PROCEDURE’ SECTION OF THE METHODS). 

 If RE-AIM not used in this aspect of the study please remove from the methods, or detail in your methods how applied and pick this up in the discussion. For example - How does RE_AIM relate to your 4FMA framework identified in your data analysis? THIS HAS NOW BEEN EXPANDED UPON IN SECTION 2.4, AND TAKEN UP IN THE DISCUSSION. 

Conclusions - these need to be tightened to reflect your findings. WE HAVE TIGHTENED THE CONCLUSIONS TO CLEARLY REFLECT OUR FINDINGS. 

Reviewer #1: 

Thank you for the opportunity to review this manuscript. It is a very interesting topic and good to see implementation acceptability being reported for ACP. Please find following my general, and more specific comments which I feel will make this a stronger manuscript. THE AUTHORS THANK REVIEWER #1 FOR TAKING THE TIME TO REVIEW THE MANUSCRIPT AND FOR THE VALUABLE INPUT AND RECOMMENDATIONS WHICH WE HAVE ENDEAVOURED TO FULLY INTEGRATE AND IMPLEMENT.

Overall:

Background: You explain the background in great depth and there are some very relevant points. However, there are some statements which do not appear strictly relevant to building your argument. Reducing some of these peripheral statements would make this section stronger and improve readability WE HAVE REMOVED STATEMENTS WHICH MAY BE CONSIDERED PERIPHERAL AND THUS NOT STRICTLY RELEVANT.

Quotes: Throughout you change formatting between quotes indented and quotes within text. It is off-putting for the reader. Decide which and stick with it. Very short quotes (a few words) within the text are fine. QUOTES ARE NOW FREESTANDING THROUGHOUT THE FINDINGS SECTION WITH THE EXCEPTION OF VERY SHORT QUOTES.

Discussion: This feels quite long and would benefit from greater contextualization in the wider literature. WE HAVE TIGHTENED AND CONTEXTUALISED THE DISCUSSION.

Implications: Again this feels long. The manuscript would benefit from the repetitions being cut and this section focusing on potential clinical and research impacts. You also mention health education here for the first time. Please introduce this in the Discussion and strengthen your argument with external literature. Overall you mention education five times, including in the conclusion, but this is not then mentioned in your abstract conclusions. If this is a significant implication it should be noted in the abstract. WE HAVE REMOVED REPETITIONS AND FOCUSED ON POTENTIAL CLINICAL AND RESEARCH IMPACTS.

WE HAVE INTRODUCED HEALTH EDUCATION IN THE DISCUSSION AND BROUGHT IN WIDER LITERATURE. PATIENT EDUCATION HAS ALSO NOW BEEN INCLUDED IN THE ABSTRACT.

Conclusions: This could be more clearly linked to your key messages. WE HAVE LINKED THE CONCLUSION MORE CLEARLY TO THE KEY MESSAGES.

Formatting: As PLOS ONE does not copyedit accepted manuscripts please do check over the full manuscript and figure for typographical errors such as use of semicolons and capitalisation etc. WE HAVE ENDEAVOURED TO IDENTIFY AND REMOVE ALL TYPOGRAPHICAL ERRORS.

Specific:

Line 10: I would have liked to see the main aim reflected here. THE MAIN AIM OF THIS PAPER IS TO REPORT ON PATIENT ACCEPTABILITY OF THE PRIMAY CARE BASED ACP INTERVENTION. THIS IS NOW MORE CLEARLY REFLECTED HERE.

Lines 17-18: This is rather confusing. Is that four practices in the control and four intervention? Or four patients within each of the eight practices to control and four patients within each of the eight practices to intervention? Neither adds up to the n=34 you interviewed. I make 8 practices and 8 participants at each to be a total of 64, half of which, the intervention group, would be 32. Obviously you know how many people you interviewed so I must have misunderstood something in the way the numbers are explained. This needs clarifying throughout the manuscript. WE HAVE NOW CLARIFIED NUMBERS OF PRACTICES AND PARTICIPANTS BOTH IN THE ABSTRACT AND IN THE METHODS SECTION. 

Line 25-26: I do not understand the relevance of the hyphens here or later in the manuscript (e.g. Line 111) THE HYPHENS HAVE BEEN REMOVED. 

Line 26-28: These are not articulated the same as the main headings in your findings section (Lines 161-163). Please correct. THIS HAS NOW BEEN CORRECTED.

Line 98: Can you add that participants gave informed consent here? I appreciate you say it below, but not to see it here raises a red flag to the reader. THIS HAS BEEN ADDED. 

Lines 105-106: See Lines 17-18 comment above. IN LINE WITH THE REQUIREMENTS OF THIS REVIEW THIS INFORMATION HAS NOW BEEN MOVED TO THE ‘SAMPLE’ SECTION, HOWEVER, IT HAS BEEN AMENDED IN ORDER TO CLARIFY NUMBERS, AND TO BRING IT IN LINE WITH THE AMENDED ABSTRACT.

Line 109: Move the bracketed element after 'for inclusion'. Also not sure you need the brackets, just a separate sentence. It would make this clearer that there is more data if the reader wishes. THE BRACKETS HAVE BEEN REPLACED BY A SEPARATE SENTENCE.

Line 110: How many met the inclusion criteria? Why did you decide on only 8 from each practice? What is your rationale? 73 PATIENTS MET THE INCLUSION CRITERIA. DETAILS ON RATIONALE REGARDING SAMPLING HAVE NOW BEEN PROVIDED.

Line 112: What was the dose? Did some only receive one? Why didn't everyone receive the same dose? Specifying would help the reader decide the weight of the findings. THIS DETAIL IS NOW INCLUDED IN SECTION 2.2: THE INTERVENTION, IN THE METHOD SECTION. NUMBER OF HOME VISITS DEPENDED ON COMPLEXITY OF NEEDS.

Line 115: Specify this was the nurse. 'They' can be confusing and imply it was the team of GP, pharmacist and nurse. If it was the team, this needs to be clarified throughout as previously it has implied it was the nurse that constructed the plan. ‘THEY’ HAS NOW BEEN REPLACED WITH ‘THE NURSES’ HERE.

Line 121: Why face to face? THIS WAS AN ERROR AND SHOULD HAVE READ “… WERE SELECTED TO TAKE PART IN A FACE-TO-FACE INTERVIEW.” WE HAVE NOW REMOVED THE EXPRESSION ALTOGETHER. 

Line 131: Can you reference some of this literature? WE HAVE NOW RECTIFIED THE OMISSION AND INCLUDED REFERENCES FOR THE RE-AIM FRAMEWORK.

Line 132: Did you use PPI? If not, it may be useful to explain your justification for this. YES, PPI WAS USED. THIS HAS NOW BEEN INCLUDED HERE.

Line 144: When? Also January 2019? YES, POST ETHICAL APPROVAL AND PRIOR TO INDIVIDUAL BASELINE DATA COLLECTION. BASELINE WAS STAGGERED OVER SEVERAL MONTHS BETWEEN FEBRUARY AND JUNE 2019.

Line 148: Who conducted the data analysis. All members of the research team? Is the research team the writing team? THE FIRST AUTHOR IN COLLABORATION WITH ANOTHER MEMBER OF THE RESEARCH TEAM (PAGE 11). THE INTERVENTION NURSES ARE NOT PART OF THE WRITING TEAM.

Line 165: Make it clearer that this is a conceptual framework which you have developed and give more details. Also, potentially mention as an output in your abstract. WE HAVE NOW CLARIFIED THAT THIS IS A NEW CONCEPTUAL FRAMEWORK WHICH WE HAVE DEVELOPED AND PROVIDED A LITTLE MORE DETAIL. WE HAVE ALSO INCLUDED THE FRAMEWORK IN THE ABSTRACT.

Line 172-176: Describing what is in the chart doesn't add anything. Try and give a high level summary for this theme here. WE HAVE EDITED THIS PARAGRAPH TO REFLECT THE SUGGESTED CHANGES.

Line 194: Where is the evidence for this? EVIDENCED THROUGH PARTICIPANT FEEDBACK DURING INTERVIEWS, AND ILLUSTRATED THROUGH THE QUOTES THROUGHOUT THE FINDINGS SECTION. THIS HAS NOW BEEN STATED IN TEXT. 

Line 226: Impossible is a very strong word. Challenging? Difficult? THIS HAS BEEN CHANGED TO ‘CHALLENGING’.

Line 268-271: I find this statement very uncomfortable. How did you analyse negative and positive mindsets? I appreciate participants gave what appeared to be two different views. Maybe report on that rather than your interpretation of what mindset that meant they had. THIS SECTION HAS NOW BEEN RE-WRITTEN IN LINE WITH THE REVIEWER’S SUGGESTION.

Line 277: This is the first mention of frailty. Are you diagnosing frailty as a syndrome, if so, how, or saying some patients were more susceptible to functional decline? I would change this title or quantify. Perhaps Impact of (fluctuating?) mental and physical capacity? WE HAVE CHANGED THE TITLE OF THIS SECTION AS PER YOUR SUGGESTION.

Line 282: Was capacity one of your inclusion criteria? If not did that impact the intervention and your study? You do not mention this in your limitations or elsewhere. CAPACITY WAS NOT AN INCLUSION CRITERION PER SE BUT THE ABILITY TO COMPLETE AN ENGLISH LANGUAGE POSTAL QUESTIONNAIRE. THIS IS NOW INCLUDED IN THE SAMPLE SECTION. 

Line 283-284: This sentence is confusing to me. (The following section makes perfect sense.) I am unsure what this togetherness of physical and mental capacity means, or where your evidence is to clarify it. WE HAVE REMOVED THIS SENTENCE.

Line 324: The use of the word subsequent makes the sentence somewhat confusing. WE HAVE REMOVED THE WORD ‘SUBSEQUENTLY’.

Line 337 and 505: Again, please clarify what you mean by the term frailty in this context. THE WORD FRAIL HAS BEEN REPLACED WITH ‘INFORM’ IN THESE SENTENCES [WHAT WAS MEANT WITH ‘FRAIL COHORT’ WAS THAT PARTICIPANTS IN THIS STUDY HAD A PRISMA-7 FRAILTY SCORE OF 3+.]

Line 357: Is spouse the right word when your example is regarding adult children? ‘SPOUSE’ WAS NOW REPLACED WITH ‘FAMILY MEMBER’.

Line 366-368: Can you evidence this with a quote? A SUPPORTING QUOTE HAS NOW BEEN INCLUDED HERE.

Reviewer #2: I think this is a useful study to examine the effects of nurse-led Anticipatory care planning.

I did not understand the following points, so I would like to ask for additional explanation. THE AUTHORS THANK REVIEWER #2 FOR TAKING THE TIME TO REVIEW THE MANUSCRIPT AND FOR THE VALUABLE INPUT AND RECOMMENDATIONS WHICH WE HAVE ENDEAVOURED TO FULLY INTEGRATE AND IMPLEMENT. WE HAVE RESPONDED TO THE REQUEST FOR ADDITIONAL EXPLANATIONS BELOW.

In the Intervention part, it is stated that the GP was divided into the intervention group and the control group by Feasibility RCT, but if this study is a cluster RCT design, please describe the flow chart of the participants. If it's not a cluster RCT design, shouldn't this description be included in the sample? PARTICIPANT RECRUITMENT HAS NOW BEEN DETAILED IN THE SAMPLE SECTION.

In addition, although it is divided into an intervention group and a control group, please describe the correspondence to the control group. WE HAVE NOW INCLUDED THIS DETAIL IN THE SAMPLE SECTION: PARTICIPANTS WERE CONTACTED BY A MEMBER OF THE RESEARCH TEAM TO INFORM THEM OF THEIR ALLOCATION TO THE CONTROL VERSUS THE INTERVENTION GROUP AFTER CONSENT HAD BEEN OBTAINED AND BASELINE STANDARDISED INTERVIEW COMPLETED. 

If this study is cluster RCT design, I think the sample size is small. THIS IS THE RECOMMENDED SAMPLE SIZE FOR FEASIBILITY PURPOSES AND IS DETAILED IN OUR PROTOCOL.

Please specify the meaning and effect of collecting data separately for the intervention group and the control group. THIS PAPER REPORTS ON THE PATIENT ACCEPTABILITY OF THE ACP INTERVENTION. TO ELICIT THIS, INTERVENTION PARTICIPANTS WERE INTERVIEWED REGARDING THEIR VIEWS AND PERCEPTIONS OF THE INTERVENTION. NO SUCH QUALITATIVE DATA WERE COLLECTED FROM THE CONTROL GROUP WHO DID NOT RECEIVE THE INTERVENTION.

QUANTITATIVE DATA WAS COLLECTED SIMULTANEOUSLY FROM INTERVENTION AND CONTROL PARTICIPANTS, HOWEVER, THIS WILL BE THE TOPIC OF ANOTHER PAPER. 

N/A

WE HAVE UPLOADED FIG 1 TO PACE AND APPENDED THE RESULTING DOCUMENT WITH OUR RESUBMISSION.

---

## [Decision Letter · Decision Letter 1]

10 Mar 2021

PONE-D-20-30516R1

Acceptability of a nurse-led, person-centred, anticipatory care planning intervention for older people at risk of functional decline: a qualitative study

PLOS ONE

Dear Dr. Corry, 

Thank you for submitting your manuscript to PLOS ONE. After careful consideration, we feel that it has merit but does not fully meet PLOS ONE’s publication criteria as it currently stands. Therefore, we invite you to submit a revised version of the manuscript that addresses the points raised during the review process.

ACADEMIC EDITOR:

Thank you for your careful consideration and response to the initial peer review comments. We have reviewed your response and detailed minor revisions required for your manuscript to meet publication standard required for PLOS ONE. Please review and respond to the editor and peer review comments below. 

We look forward to receiving your revised manuscript.

Kind regards,

Catherine J Evans, PhD, MSc, BSc (Hons)

Academic Editor

PLOS ONE

Journal Requirements:

Additional Editor Comments (if provided):

Please use language of frail rather than infirm throughout. (see #Reviewer 1 point below)

Methods

Detail on the RCT needs to be in the design and procedure section including reference for the published protocol. This then details the design at the beginning of the methods for the reader to understand the intervention and qualitative interviews are part of a trial.

Reporting follows COREQ guidance. This needs to be detailed in the design section as relates to the whole study reporting, not only the intervention. Can you move up from the intervention detail and report in the design subsection.

Sample

Can you revise this for clarity – moving information on the trial design to the study design section.

State clearly at the beginning sample eligibility, then give the detail. Such as ..

The qualitative interviews included all participants allocated to the intervention group in the feasibility cluster RCT. Detailing on the sampling method for the trial can remain in this sub-section.

Table 1 – can you include this detailed table as a supplementary file, and cite in the manuscript as supplementary. It is important information, but is not main results. Include  in the results  table 1 reporting characteristics of the qual sample (see below).

Line 156 – please remove pre-COVID, not needed as implicit from the dates given that data collection occurred before the 2020 pandemic

Interview schedule - line 164 - please state how PPI supported the study and give an example to illustrate contribution to the interview schedule. Please detail if a PPI group - number, how worked with them referring to GRIPP reporting for PPI.

Findings

Can you present in the results a table 1 summary table of the qualitive interview participant characteristics e.g. Age median (range IQR), gender – proportions, average PRISMA score i.e.d median and spread, jurisdiction – proportions, allocated group etc. You can then reduce the narrative reporting on the participant results to main points and refer to the table 1.

Line 210 – please remove detail on methods of data analysis. This is repeating detail given in the methods. Please report your findings to indicated what found – past tense. Not we sought to elicit – future tense as you are reporting findings. Reporting on the themes and framework begin as a new para, Line 209 new para to report the themes identified. State suggestion below, then give the detail re the themes and the framework

Our findings on the acceptability of the intervention formed five main themes:

Use of the word correct – review 1, suggested use appropriate timing. I agree with this – from your findings this is about the appropriate time for the person. Not a correct time or un-correct time which implies a biomedical perspective of disease stage. This is about the priorities for the person, not their disease stage.

Discussion

Limitations – line 611 – no cross jurisdictional differences, but the sample all identified as ethnically White. Transferability is limited to ethnically White. Please indicate in the limitations this point sample ethnically White. This limits transferability to Black and Minority ethnic groups.

Reviewers' comments:

Reviewer's Responses to Questions

**Comments to the Author**

1. If the authors have adequately addressed your comments raised in a previous round of review and you feel that this manuscript is now acceptable for publication, you may indicate that here to bypass the “Comments to the Author” section, enter your conflict of interest statement in the “Confidential to Editor” section, and submit your "Accept" recommendation.

Reviewer #1: (No Response)

Reviewer #2: All comments have been addressed

2. Is the manuscript technically sound, and do the data support the conclusions?

Reviewer #1: Yes

Reviewer #2: Yes

3. Has the statistical analysis been performed appropriately and rigorously? 

Reviewer #1: N/A

Reviewer #2: N/A

4. Have the authors made all data underlying the findings in their manuscript fully available?

Reviewer #1: Yes

Reviewer #2: Yes

5. Is the manuscript presented in an intelligible fashion and written in standard English?

Reviewer #1: Yes

Reviewer #2: Yes

6. Review Comments to the Author

Reviewer #1: Thank you for the opportunity to review the amended manuscript. It is, I feel, far clearer and stronger. I do have some additional comments which I believe will strengthen the manuscript further:

Throughout - I assume comments from the author team have been answered to everyone's satisfaction? e.g. lines 709 & 713 on the annotated copy.

Throughout - Changing frail to infirm and changing the heading. This is perhaps a misunderstanding of my comment. I was concerned you had categorized frailty without explaining what that meant to you within the paper. You have now explained PRISMA-7 and its role in flagging frailty well and so may I ask that you go back to using the term frail, or older people living with frailty, or frail elders. Infirm has its own negative connotations.

Throughout - Appropriate rather than 'correct' timing.

Line 103 - The reader needs to know a bit more contextualisation. "at the same time as collecting time 2 quant data" isn't enough.

Lines 160 - 174 - level of detail is much better, but perhaps the actual questions could sit in a separate table/box so the reader doesn't get lost now you've added the aim of the questions. Also, may need to explain why you are using advance CP and anticipatory CP here.

Line 163-164 - could you mention briefly how the PPI supported this?

Line 187 - much clearer. Can you add the initials of the writing team member here?

Line 589 - 590 - Could you expand a little on patient education and appropriate timing, and link this to the wider literature?

Reviewer #2: Thank you for revise your paper. I think it's a very useful paper. Because you had revision it carefully, I was able to understand. I am waiting for a research paper with other data to be published.

7. PLOS authors have the option to publish the peer review history of their article (what does this mean?). If published, this will include your full peer review and any attached files.

Reviewer #1: No

Reviewer #2: No

---

## [Author Response · Author response to Decision Letter 1]

24 Mar 2021

Response to Reviewers 19/03/2021

Once again, the authors would like to express their sincere gratitude to the editors and reviewers for giving of their valuable time to review our resubmission and providing further helpful suggestions for revision, all of which we have endeavoured to take on board, implement and incorporate, thus making it a stronger manuscript. We have addressed each point raised during the second review and responded to it with the relevant remedial action and reply as indicated in the table below.

Points raised by Editors and Reviewers Authors’ response and action

Journal Requirements:

WE HAVE REVIEWED THE REFERENCES FOR COMPLETION AND CORRECTNESS.

WE HAVE ADDED TWO MORE REFERENCES (36 AND 50) AND ADJUSTED NUMBERING THROUGHOUT. 

Additional Editor Comments (if provided):

 THE AUTHORS THANK THE EDITOR FOR REVIEWING THE AMENDED MANUSCRIPT AND SUGGESTIONS TO FURTHER STRENGTHEN IT. WE HAVE ENDEAVOURED TO RESPOND TO EACH POINT AS BELOW.

Please use language of frail rather than infirm throughout. (see #Reviewer 1 point below) ‘INFIRM’ WAS REPLACED WITH ‘FRAIL’ THROUGHOUT (2 REPLACEMENTS MADE).

Methods

Detail on the RCT needs to be in the design and procedure section including reference for the published protocol. This then details the design at the beginning of the methods for the reader to understand the intervention and qualitative interviews are part of a trial.

 DETAILS ABOUT THE INTERVENTION, INCLUDING REFERENCE TO THE PUBLISHED PROTOCOL HAVE NOW BEEN MOVED TO THE BEGINNING OF THE ‘DESIGN AND PROCEDURE’ SECTION IN THE METHODS. THE PARAGRAPH ABOUT THE QUALITATIVE INTERVIEWS IMMEDIATELY FOLLOWS THIS.

Reporting follows COREQ guidance. This needs to be detailed in the design section as relates to the whole study reporting, not only the intervention. Can you move up from the intervention detail and report in the design subsection.

 THE SENTENCE: ‘THIS PAPER REPORTS ON PATIENT ACCEPTABILITY OF THE INTERVENTION, AND FOLLOWS THE COREQ GUIDELINES FOR REPORTING QUALITATIVE RESEARCH [27] (SEE S3) AND THE TIDIER CHECKLIST [28] (SEE S4)’ HAS NOW BEEN MOVED TO SECTION 2.1 ‘DESIGN AND PROCEDURE’. 

Sample

Can you revise this for clarity – moving information on the trial design to the study design section.

State clearly at the beginning sample eligibility, then give the detail. Such as .. THE SENTENCE PERTAINING TO TRIAL DESIGN HAS BEEN MOVED TO THE STUDY DESIGN SECTION. THE SENTENCE DENOTING INCLUSION CRITERIA HAS BEEN MOVED TO LEAD ON FROM DESCRIPTION OF ELIGIBLITY AND INCLUSION (LINES 136-139).

The qualitative interviews included all participants allocated to the intervention group in the feasibility cluster RCT. Detailing on the sampling method for the trial can remain in this sub-section. THIS INFORMATION REMAINED IN THIS SUB-SECTION.

Table 1 – can you include this detailed table as a supplementary file, and cite in the manuscript as supplementary. It is important information, but is not main results. Include in the results table 1 reporting characteristics of the qual sample (see below).

 THIS TABLE IS NOW INCLUDED AS SUPPLEMENTARY FILE 5 (S5): TABLE 2. PARTICIPANT CHARACTERISTICS, AND IS REFERRED TO AS SUCH IN THE MANUSCRIPT. 

Line 156 – please remove pre-COVID, not needed as implicit from the dates given that data collection occurred before the 2020 pandemic

 THE PHRASE ‘PRE-COVID’ HAS BEEN REMOVED HERE.

Interview schedule - line 164 - please state how PPI supported the study and give an example to illustrate contribution to the interview schedule. Please detail if a PPI group - number, how worked with them referring to GRIPP reporting for PPI. THE FOLLOWING SENTENCES HAVE NOW BEEN INSERTED HERE:

FOLLOWING GRIPP GUIDELINES ON PPI REPORTING [36] WE ENGAGED THREE PPI (ONE IN ROI, TWO IN NI) IN AN ADVISORY CAPACITY TO ATTEND REGULAR PROJECT TEAM MEETINGS AND TO DISCUSS PROTOCOL, ANALYSIS, AND DISSEMINATION, AS WELL AS PROGRESSION AND NEXT STEPS, SPECIFICALLY TO CONSULT ON STUDY DOCUMENTS, INCLUDING QUALITATIVE AND QUANTITATIVE INTERVIEW SCHEDULES, TO ENSURE INCORPORATION OF THE VITAL LAY PERSON PERSPECTIVE. AN EXAMPLE OF PPI INPUT TO THE QUALITATIVE INTERVIEW SCHEDULE IS THE CHANGE FROM ‘DID YOU FEEL ACTIVELY INVOLVED IN YOUR DISCUSSIONS WITH THE NURSE TO IDENTIFY YOUR HEALTHCARE NEEDS?’ TO ‘DID YOU HAVE ENOUGH INPUT IN IDENTIFYING YOUR HEALTH NEEDS AND DEVELOPING YOUR CARE PLAN?’

Findings

Can you present in the results a table 1 summary table of the qualitive interview participant characteristics e.g. Age median (range IQR), gender – proportions, average PRISMA score i.e.d median and spread, jurisdiction – proportions, allocated group etc. You can then reduce the narrative reporting on the participant results to main points and refer to the table 1. PREVIOUS TABLE 1 WITH DETAILED PARTICIPANT DETAILS HAS NOW BEEN INCLUDED AS A SUPPLEMENTARY FILE TABLE 2 (S5) AS REQUESTED ABOVE. WE HAVE NOW INCLUDED A SUMMARY TABLE WITHIN THE TEXT (TABLE 1), SHOWING AGE MEAN AND SD, GENDER, PRISMA SCORE MEAN AND SD, AND DISTRIBUTIONS OF ROI/NI, INTERVENTION/ CONTROL, AND URBAN/ RURAL. 

Line 210 – please remove detail on methods of data analysis. This is repeating detail given in the methods. Please report your findings to indicated what found – past tense. Not we sought to elicit – future tense as you are reporting findings.

 Reporting on the themes and framework begin as a new para, Line 209 new para to report the themes identified. State suggestion below, then give the detail re the themes and the framework

Our findings on the acceptability of the intervention formed five main themes:

Use of the word correct – review 1, suggested use appropriate timing. I agree with this – from your findings this is about the appropriate time for the person. Not a correct time or un-correct time which implies a biomedical perspective of disease stage. This is about the priorities for the person, not their disease stage. DETAIL ON METHODS OF ANALYSIS HAVE BEEN REMOVED.

WE HAVE INSERTED A PARAGRAPH AT LINE 222 AND STARTED SENTENCE IN PAST TENSE (‘ANALYSIS OF INTERVIEW DATA RESULTED IN FIVE MAIN THEMES …’), OMITTING THE PREVIOUS ‘WE SOUGHT TO ELICIT’. 

THE WORD ‘CORRECT’ HAS BEEN REPLACED BY ‘APPROPRIATE’ IN RELATION TO TIMING THROUGHOUT.

Discussion

Limitations – line 611 – no cross jurisdictional differences, but the sample all identified as ethnically White. Transferability is limited to ethnically White. Please indicate in the limitations this point sample ethnically White. This limits transferability to Black and Minority ethnic groups. IN LIMITATIONS, LINE 638 NOW READS: ‘AS THE SAMPLE WAS ENTIRELY ETHNICALLY WHITE TRANSFERABILITY TO OTHER ETHNIC GROUPS MAY BE LIMITED.’

Reviewer #1: 

Thank you for the opportunity to review the amended manuscript. It is, I feel, far clearer and stronger. I do have some additional comments which I believe will strengthen the manuscript further: THE AUTHORS THANK REVIEWER #1 FOR REVIEWING THE AMENDED MANUSCRIPT AND SUGGESTIONS TO FURTHER STRENGTHEN IT. WE HAVE ENDEAVOURED TO RESPOND TO EACH POINT AS BELOW. 

Throughout - I assume comments from the author team have been answered to everyone's satisfaction? e.g. lines 709 & 713 on the annotated copy.

 APOLOGIES; IT IS UNCLEAR WHAT THIS POINT IS REFERRING TO? LINES 709 AND 713 OF THE MANUSCRIPT ARE IN THE REFERENCE SECTION. WE ARE HAPPY TO ADDRESS THIS ONCE CLARIFIED. 

Throughout - Changing frail to infirm and changing the heading. This is perhaps a misunderstanding of my comment. I was concerned you had categorized frailty without explaining what that meant to you within the paper. You have now explained PRISMA-7 and its role in flagging frailty well and so may I ask that you go back to using the term frail, or older people living with frailty, or frail elders. Infirm has its own negative connotations. WE HAVE CHANGED THE TERM ‘INFIRM’ BACK TO ‘FRAIL’ THROUGHOUT THE MANUSCRIPT. 

THE HEADING HAS NOT CONTAINED EITHER OF THESE TERMS BUT REFLECTS THE WORDING OF THE STUDY TITLE WITH THE TERM ‘AT RISK OF FUNCTIONAL DECLINE’ WHICH IS USED THROUGHOUT THE MANUSCRIPT AS WELL. 

Throughout - Appropriate rather than 'correct' timing. THIS HAS BEEN AMENDED THROUGHOUT THE MANUSCRIPT (2 INCIDENTS).

Line 103 - The reader needs to know a bit more contextualisation. "at the same time as collecting time 2 quant data" isn't enough. THIS SENTENCE (STARTING AT LINE 121) READS NOW: THIS INVOLVED QUALITATIVE INTERVIEWS WITH PARTICIPANTS IN THEIR OWN HOMES AT 10-WEEK FOLLOW-UP (AUGUST TO OCTOBER 2019) FOLLOWING COMPLETION OF THE INTERVENTION, DURING THE SAME VISIT AT WHICH TIME 2 QUANTITATIVE DATA WERE COLLECTED. PARTICIPANTS COMPLETED THE QUANTITATIVE QUESTIONNAIRE WITH THE RESEARCHER AND WERE THEN INTERVIEWED.

Lines 160 - 174 - level of detail is much better, but perhaps the actual questions could sit in a separate table/box so the reader doesn't get lost now you've added the aim of the questions. Also, may need to explain why you are using advance CP and anticipatory CP here. THE QUESTIONS HAVE NOW BEEN PLACED IN A BOX WITHIN THE TEXT.

WE HAVE REPLACED THE MENTION OF ‘ADVANCE CARE PLANNING’ WITH ‘ANTICIPATORY CARE PLANNING’. 

Line 163-164 - could you mention briefly how the PPI supported this?

 THE FOLLOWING SENTENCE HAS NOW BEEN INSERTED HERE (STARTING AT LINE 166): 

FOLLOWING GRIPP GUIDELINES ON PPI REPORTING [36] WE ENGAGED THREE PPI IN AN ADVISORY CAPACITY TO ATTEND REGULAR PROJECT TEAM MEETINGS AND TO DISCUSS PROGRESSION, NEXT STEPS, AND CONSULT ON STUDY DOCUMENTS, INCLUDING QUALITATIVE AND QUANTITATIVE INTERVIEW SCHEDULES, TO ENSURE INCORPORATION OF THE VITAL LAY PERSON PERSPECTIVE. AN EXAMPLE OF PPI INPUT TO THE QUALITATIVE INTERVIEW SCHEDULE IS THE CHANGE FROM ‘DID YOU FEEL ACTIVELY INVOLVED IN YOUR DISCUSSIONS WITH THE NURSE TO IDENTIFY YOUR HEALTHCARE NEEDS?’ TO ‘DID YOU HAVE ENOUGH INPUT IN IDENTIFYING YOUR HEALTH NEEDS AND DEVELOPING YOUR CARE PLAN?’

Line 187 - much clearer. Can you add the initials of the writing team member here? THE INITIALS OF THE TEAM MEMBER – KB – HAVE NOW BEEN INSERTED HERE.

Line 589 - 590 - Could you expand a little on patient education and appropriate timing, and link this to the wider literature? LINES 601-604 NOW READ: HEALTH EDUCATION FOR OLDER ADULTS CAN BE VERY EFFECTIVE, BOTH IN TERMS OF IMPROVING INTERVENTION ADHERENCE AND POTENTIALLY IN REDUCING MORBIDITY AND EXCESS MORTALITY [40,50]. IT COULD HELP IMPROVE HEALTH LITERACY [39,42] AND ENSURE KNOWLEDGE AND UNDERSTANDING OF ACP, THUS FACILITATING TIMELY UPTAKE.

Reviewer #2: 

Thank you for revise your paper. I think it's a very useful paper. Because you had revision it carefully, I was able to understand. I am waiting for a research paper with other data to be published. THE AUTHORS THANK REVIEWER #2 FOR THEIR REVIEW AND COMMENDATION. 

WE CURRENTLY HAVE ANOTHER QUALITATIVE PAPER UNDER REVIEW, AND ONE IN PREPARATION. WE ARE ALSO PREPARING A PAPER REPORTING ON ALL QUANTITATIVE ASPECTS OF THE STUDY, INCLUDING A HEALTH ECONOMIC EVALUATION.

AUTHORS’ CHANGE

ACKNOWLEDGEMENTS WE HAVE ALTERED THE SENTENCE RECOGNISING THE NICRN CONTRIBUTION TO BE A MORE SPECIFIC ACKNOWLEDGEMENT WHICH NOW READS:

‘WE THANK THE NORTHERN IRELAND CLINICAL RESEARCH NETWORK [PRIMARY CARE] (NICRN PC) FOR THE RECRUITMENT OF GP PRACTICES AND THE DELIVERY OF THE ACP INTERVENTION IN NORTHERN IRELAND.’

---

## [Editor Report · Decision Letter 2]

7 Apr 2021

PONE-D-20-30516R2

Acceptability of a nurse-led, person-centred, anticipatory care planning intervention for older people at risk of functional decline: a qualitative study

PLOS ONE

Dear Dr. Corry, 

Thank you for submitting your manuscript to PLOS ONE. After careful consideration, we feel that it has merit but does not fully meet PLOS ONE’s publication criteria as it currently stands. Therefore, we invite you to submit a revised version of the manuscript that addresses the points raised during the review process.

ACADEMIC EDITOR: Please see my detailed comments below. Key points are to revise table 1, move table 2 to supplementary and remove supplementary files reporting the full transcripts. Please reads detailed comments below and respond in full. 

We look forward to receiving your revised manuscript.

Kind regards,

Catherine J Evans, PhD, MSc, BSc (Hons)

Academic Editor

PLOS ONE

Journal Requirements:

Additional Editor Comments (if provided):

Thank you for carefully reviewing the peer review and editor comments, and amending the manuscript accordingly. Most comments have been responded to. However, as editor I have comments below. Please respond to these in full.

Key words

Please review to ensure using MeSH terms to describe the study – the population, focus, methods. Examples of potential MeSH terms:

Frailty; Primary health care; Qualitative Research; Randomised Controlled Trial

Line 123 – please reviewing phrasing to improve clarity

During the visit, quantitative data was also collected. Participants completed the quantitative questionnaires for timepoint two, then the qualitative interview.

Line 126 – can you state the interview length range and the measure of average – presume median average of 60 minutes (range x to y)

Line 128 – is the experienced female researcher one of the co-authors? If yes, please insert the initials

Line 128 - Hospital admission is this any admission, or unplanned only? If any admission leave as is, if unplanned only please state unplanned hospital admission

Table 1 – this is reporting the results. Table 1 reporting the participants should be in the results section. Please can you report the characteristics of the sample in line with your stated inclusion criteria. Detail a row for each inclusion criteria with the measure clearly stated such as, Mean (SD), if n state n= . The reader needs to be able to assess the extent your sample reflects the sampling criteria, and understand the characteristics of the participants. Please look at publications in PLOS One and reporting style for participant characteristics, typically reported in a table. You report in the results characteristics of the participants by the respective jurisdiction. If this distinction is important, report this in table 1 using a column for each jurisdiction to report the respective characteristic. Or if not important, report as a single sample.

You want to use table 1 to convey the important characteristics of your sample relevant to your findings and conclusions drawn. Lines 212-222 –this data should be reported in your table 1 e.g. living status, marital status. In the narrative reporting of the results, report main points important to emphasize, such factors increasing risk of functional decline like frailty, average age, number of conditions, polypharmacy, number of unplanned hospital admissions in the past year. Only one partner present – is an important point. Add to the table detail re identified family carer – and who they were. This is information is reported in section 3.3.2. Reporting in the table would improve clarity and give emphasis, particularly as care partners is a main theme and essential for readers to understand extent the participants identified an family carer or not to appraise critically findings on the importance of family carers. Care partners – this is an unusual way to describe family carers or informal carers. Can you state family carers (including close friends) and use term carers or caregivers throughout . In figure 1, state family carers – this would improve clarity as to who is being referred too.

Table 1 should state Average number of medications as medication review a component of the intervention and a theme identified re acceptability. Important for the reader to understand extent of polypharmacy and relevance of medication

review

Table 2 is the raw data – this needs to be moved to supplementary. 

Lines 150-161 is long giving detail on the trial. Please reduce and keeping reporting on methods relevant to the paper reporting the qualitative interview studies. Please state Out of 73 patients meeting eligibility, 64 were recruited and allocated to intervention (n=32) or control (n=32) group after consent and baseline data collection. All patients in the intervention group were invited to complete a qualitative interview at 10-week follow-up…... Detail on variation in recruitment by GP practice does not seem relevant to reporting in this paper about the qualitative interview findings. Detail on consent is reported in the ethical considerations.

Line 179 – topic guides, items included the following. Please either list or put in a figure, and indicate in the text end of line 178 (see figure 1)

Line 185 – move up to line 174 to keep reporting on the interview guide together. State all interviews were conducted between August to October 2019. Please remove sentence re unaffected by coronavirus pandemic. The sentence has no relevance to the study.

Supplementary files – all the transcripts are made available. This seems ethically inappropriate as I am assuming the participants consented for publication of quotes, not the whole transcript with potential for them to be identified by e.g. their GP, family members. Please remove the transcripts as supplementary and indicate data is available with appropriate ethical approvals from the authors. You can deposit the raw data in a national repository for qualitative data. This then ensures the research governance and management to the data, such as destroyed after stated time as indicated in your ethical approval.

---

## [Author Response · Author response to Decision Letter 2]

5 May 2021

Response to Reviewers 16 April, 2021

The authors would like to express their sincere gratitude to the Editor for giving of her valuable time to review our resubmission and providing further helpful suggestions for revision, all of which we have endeavoured to take on board, implement and incorporate, thus making it a stronger manuscript. We have addressed each point raised during the last review and responded to it with the relevant remedial action and reply as indicated in the table below.

Points raised by Editors and Reviewers Authors’ response and action

Key words

Please review to ensure using MeSH terms to describe the study – the population, focus, methods. Examples of potential MeSH terms:

Frailty; Primary health care; Qualitative Research; Randomised Controlled Trial

We have now used MeSH on Demand (nih.gov) in conjunction with the Editor’s suggestions for Keywords. They are now:

Keywords: older adults, frailty, anticipatory care planning, primary health care, qualitative research, randomised controlled trial, UK, Republic of Ireland

Line 123 – please review phrasing to improve clarity. During the visit, quantitative data was also collected. Participants completed the quantitative questionnaires for timepoint two, then the qualitative interview. Lines 122-123 now read: ‘During the visit quantitative data was also collected. Participants completed the quantitative questionnaires for the 10-week follow-up, then the qualitative interview.’

Line 126 – can you state the interview length range and the measure of average – presume median average of 60 minutes (range x to y)

 Line 125 now states the correct length and median average for the qualitative interview: ‘The qualitative interviews had a median average length of nine minutes (range: three to 24 minutes)’ …

The Authors thank the Editor for highlighting this point as the 60 minutes previously stated were inclusive of the quantitative data collection.

Line 128 – is the experienced female researcher one of the co-authors? If yes, please insert the initials

 Line 127 - Yes, she is, and her initials have now been inserted in the sentence: ‘The interviews were conducted by an experienced female researcher (DC) who had not met the participants prior to interview.’ 

Line 128 - Hospital admission is this any admission, or unplanned only? If any admission leave as is, if unplanned only please state unplanned hospital admission

 Line 136 – ‘hospital admission’ refers to any admission; sentence was left as is.

Table 1 – this is reporting the results. Table 1 reporting the participants should be in the results section. 

Please can you report the characteristics of the sample in line with your stated inclusion criteria. Detail a row for each inclusion criteria with the measure clearly stated such as, Mean (SD), if n state n= . The reader needs to be able to assess the extent your sample reflects the sampling criteria, and understand the characteristics of the participants. 

Please look at publications in PLOS One and reporting style for participant characteristics, typically reported in a table. 

You report in the results characteristics of the participants by the respective jurisdiction. If this distinction is important, report this in table 1 using a column for each jurisdiction to report the respective characteristic. Or if not important, report as a single sample.

 Table 1 has been moved to the results section. 

The characteristics of the sample have been reported in line with our stated inclusion criteria (a row for each*), with the addition of gender, marital status & living arrangements; and urbanicity, and family carer participation. Means (SD) and n have been provided where applicable. 

We have inspected a number of publications in PLOS ONE and reporting style (table) for participant characteristics and have applied it to Table 1.

The distinction is unimportant in the context of this paper, and the sample was homogenous across jurisdictions, therefore characteristics are reported as a single sample. 

* While patient health conditions 

were recorded in the nurses’ clinical record, this information was not considered relevant to the study objectives thus not abstracted and analysed for research purposes.

You want to use table 1 to convey the important characteristics of your sample relevant to your findings and conclusions drawn. Lines 212-222 –this data should be reported in your table 1 e.g. living status, marital status.

In the narrative reporting of the results, report main points important to emphasize, such factors increasing risk of functional decline like frailty, average age, number of conditions, polypharmacy, number of unplanned hospital admissions in the past year. 

Only one partner present – is an important point. Add to the table detail re identified family carer – and who they were. 

This is information is reported in section 3.3.2. Reporting in the table would improve clarity and give emphasis, particularly as care partners is a main theme and essential for readers to understand extent the participants identified a family carer or not to appraise critically findings on the importance of family carers.

Care partners – this is an unusual way to describe family carers or informal carers. Can you state family carers (including close friends) and use term carers or caregivers throughout. 

In figure 1, state family carers – this would improve clarity as to who is being referred too.

Living arrangements and marital status have now been included in Table 1. 

The narrative has now been edited according to the Editor’s suggestions as below (We do not have detail on planned vs unplanned hospital admissions. While patient health conditions 

were recorded in the nurses’ clinical record, this information was not considered relevant to the study objectives thus not abstracted and analysed for research purposes.):

‘The average PRISMA-7 score of 4.15 (1.12) in our sample was indicative of an increased risk of frailty and the need for further clinical review. As per inclusion criteria all participants had 2 or more chronic conditions; were taking on average 11.39 medications; had 5.2 GP visits during the past year; and an average of 6.4 inpatient nights in the previous year. Only one family carer actively took part in the interview. 

All participants were white European. Gender was evenly distributed, with a mean sample age of 80.13. The majority were married (61.8%) and lived in urban areas (58.82%).’

This family carer was present in the sense that she actively participated in the interview, unlike the other family carers who were present in the house or in the room but did not contribute to the interview. We have included this information in Table 1 now.

The term ‘care partners’ has now been replaced with ‘family carers’ throughout the manuscript and in Figure 1. We have put Figure 1 through PACE again following this change.

Table 1 should state Average number of medications as medication review a component of the intervention and a theme identified re acceptability. Important for the reader to understand extent of polypharmacy and relevance of medication

review

 We have now included the average number of medications in Table 1. 

Table 2 is the raw data – this needs to be moved to supplementary. 

 Table 2 has been removed from the manuscript and will be submitted as supplementary file S3.

Lines 150-161 is long giving detail on the trial. Please reduce and keeping reporting on methods relevant to the paper reporting the qualitative interview studies. 

Please state Out of 73 patients meeting eligibility, 64 were recruited and allocated to intervention (n=32) or control (n=32) group after consent and baseline data collection. All patients in the intervention group were invited to complete a qualitative interview at 10-week follow-up…... 

Detail on variation in recruitment by GP practice does not seem relevant to reporting in this paper about the qualitative interview findings. Detail on consent is reported in the ethical considerations. Lines 145-149 now read: ‘Out of 73 patients meeting eligibility, 65 were recruited and randomly allocated to intervention (n=34) or control (n=31) group after consent and baseline data collection. All patients in the intervention group were invited to complete a qualitative interview at 10-week follow-up (August to October, 2019) to explore the acceptability of the ACP intervention.’

All other details in this paragraph in terms of recruitment, allocation, and consent have been removed. 

Line 179 – topic guides, items included the following. Please either list or put in a figure, and indicate in the text end of line 178 (see figure 1)

 Line 167 now reads … Topic guide items included the following:

The items are now listed.

Line 185 – move up to line 174 to keep reporting on the interview guide together.

State all interviews were conducted between August to October 2019. 

Please remove sentence re unaffected by coronavirus pandemic. The sentence has no relevance to the study.

 This sentence has now been moved up and starts at line 164.

We now state that all interviews were conducted between August and October 2019.

We have removed the sentence regarding the coronavirus pandemic.

Supplementary files – all the transcripts are made available.

This seems ethically inappropriate as I am assuming the participants consented for publication of quotes, not the whole transcript with potential for them to be identified by e.g. their GP, family members. 

Please remove the transcripts as supplementary and indicate data is available with appropriate ethical approvals from the authors. You can deposit the raw data in a national repository for qualitative data. This then ensures the research governance and management to the data, such as destroyed after stated time as indicated in your ethical approval. We have removed the transcripts. (They had been added following request at first review.)

We are in agreement with the Editor. 

We have removed the supplementary files and indicated that data is available with appropriate ethical approvals from the authors.

---

## [Editor Report · Decision Letter 3]

7 May 2021

Acceptability of a nurse-led, person-centred, anticipatory care planning intervention for older people at risk of functional decline: a qualitative study

PONE-D-20-30516R3

Dear Dr. Corry, 

We’re pleased to inform you that your manuscript has been judged scientifically suitable for publication and will be formally accepted for publication once it meets all outstanding technical requirements.

Kind regards,

Catherine J Evans, PhD, MSc, BSc (Hons)

Academic Editor

PLOS ONE
---

## [Editor Report · Acceptance letter]

11 May 2021

PONE-D-20-30516R3 

Acceptability of a nurse-led, person-centred, anticipatory care planning intervention for older people at risk of functional decline: a qualitative study 

Dear Dr. Corry:

I'm pleased to inform you that your manuscript has been deemed suitable for publication in PLOS ONE. Congratulations! Your manuscript is now with our production department. 

Kind regards, 

on behalf of

Dr. Catherine J Evans 

Academic Editor

PLOS ONE